# Oxygen isotope ($\delta^{18}O$, $\Delta'^{17}O$) insights into continental mantle evolution since the Archean

Ilya N. Bindeman [1✉], Dmitri A. Ionov [2], Peter M. E. Tollan [3,4] & Alexander V. Golovin[5,6]

Oxygen isotopic ratios are largely homogenous in the bulk of Earth's mantle but are strongly fractionated near the Earth's surface, thus these are robust indicators of recycling of surface materials to the mantle. Here we document a subtle but significant ~0.2‰ temporal decrease in $\delta^{18}O$ in the shallowest continental lithospheric mantle since the Archean, no change in $\Delta'^{17}O$ is observed. Younger samples document a decrease and greater heterogeneity of $\delta^{18}O$ due to the development and progression of plate tectonics and subduction. We posit that $\delta^{18}O$ in the oldest Archean samples provides the best $\delta^{18}O$ estimate for the Earth of 5.37‰ for olivine and 5.57‰ for bulk peridotite, values that are comparable to lunar rocks as the moon did not have plate tectonics. Given the large volume of the continental lithospheric mantle, even small decreases in its $\delta^{18}O$ may explain the increasing $\delta^{18}O$ of the continental crust since oxygen is progressively redistributed by fluids between these reservoirs via high-$\delta^{18}O$ sediment accretion and low-$\delta^{18}O$ mantle in subduction zones.

[1] Department of Earth Sciences, University of Oregon, Eugene, OR, USA. [2] Department of Geosciences, Montpellier University, Montpellier, France. [3] Department of Earth Sciences, ETH Zurich, Zurich, Switzerland. [4] Gubelin Gem Lab, Lucerne, Switzerland. [5] Sobolev Institute of Geology and Mineralogy Siberian Brach RAS, Novosibirsk, Russia. [6] Institute of the Earth's Crust, Siberian Branch, RAS, Irkutsk, Russia. ✉email: bindeman@uoregon.edu

Earth's peridotitic mantle is the predominant reservoir of the planet, largely controlling elemental and isotopic mass balances of lithophilic elements and their isotopes planet-wide[1]. Furthermore, the temporal evolution of the mantle may exhibit large impacts on the lithosphere, hydrosphere, and atmosphere however this topic remains unexplored.

The formation of the mantle is due to the accretion of Earth from different types of meteorites and planetesimals at 4.56 Ga, followed by its quick segregation from the iron-rich core[2]. Perhaps the late and most significant episode of Earth's accretion was the Giant Impact at 4.44 Ga when a Mars-sized object Thea collided with a differentiated Earth to expel material in the close gravitational proximity of the Earth forming the moon[3,4]. The composition of Thea with respect to its major elements and isotopes (e.g., oxygen) was not dramatically different from that of the Earth[5,6]. The magma oceans that followed the collision further homogenized the mantles of both the Earth and the moon. Their subsequent geologic histories, however, were dramatically different as the moon is a smaller body that finished its geological evolution early and did not have plate tectonics or abundant hydrosphere that interacted with rocks on the surface.

We have only limited information about the Earth's mantle provided by basaltic magmas sampling the convective deeper mantle, all the way down to the lower core-mantle boundary layer[1], and shallow basalts and mantle fragments (xenoliths) in kimberlites that sample the uppermost boundary layer: the 100–200 km-thick continental lithospheric mantle (CLM). As Earth cools and develops thicker crust and continents, the CLM too is expected to grow and evolve.

Early Earth after crystallization of the magma ocean soon after 4.44 Ga developed its first crust at 4.4–4.3 Ga which preserves evidence of hydrosphere-lithosphere interaction as early as the Hadean[7–10]. Only zircons are available in the 4.4–4.05 Ga time frame on Earth to shed light on these early processes and the earliest continental terrestrial rocks, such as 4.05 Ga Acasta Gneiss[11]. However, 4.1–4.3 Ga basalts and zircons are available on the moon[3,4,12] and their isotopic values may provide a common point of comparison with Earth.

Oxygen is the main element on terrestrial planets. The exact O isotope composition of the Bulk Silicate Earth (BSE) is essential to understanding the Earth's accretion from different types of parental meteoritic reservoirs, including Thea, subsequent differentiation into crust and CLM, yet these processes remain to be well-constrained[1–4]. Even small $\delta^{18}O$ variations in global reservoirs are significant because oxygen is by far the most abundant chemical element on Earth. Estimates range from 5.1 to 5.9‰, depending on whether the value is derived from data on meteorites or terrestrial samples[3,4,13], while the debate on $\Delta'^{17}O$ values, centers around the initial difference and the degree of mixing between Thea, moon, and the Earth after the Giant Impact[4,5].

It is abundantly clear that the surface of the Earth was covered by oceans since the 4.4 Ga[7,8]. As the basaltic crust was largely submarine, it interacted with seawater both at low and high temperatures[9,10]. Oxygen isotopes may be the best and most straightforward tool in addressing the effects of water–rock interaction and recycling of surface-exposed materials[14,15]. In the modern world, oceanic crust, created at mid-ocean ridges (MOR), undergoes pervasive hydrothermal exchange with seawater, leading to heavy $\delta^{18}O$ (+7–13‰) at the top in pillow basalts and lower $\delta^{18}O$ (+2–5‰) at the bottom in gabbros and peridotites[14]. Olivine with higher and lower $\delta^{18}O$ than the average olivine phenocrysts from mantle-equilibrated MOR basalts has been found in many island arcs around the world[15,16] and attributed to the dehydration or melting of this altered oceanic crust during subduction. Eclogites that represent metamorphosed basalts in

subducting slabs display a similar range of $\delta^{18}O$ values, suggesting that hydrothermally altered slabs were subducted in the past at least since 2.5 Ga[15,17,18] and perhaps as early as 3.0–3.2 Ga[19], a time window when eclogitic diamonds became prevalent over peridotitic.

The higher temperature of the Hadean and early Archean mantle[2,20], leads to a higher degree of mantle melting and precludes modern-style plate tectonics from a physical point of view, with its subduction and crustal recycling driven by the transformation of basaltic crust into dense eclogites[21]. The time of initiation of plate tectonics on Earth is a topic of current debate[9–12]. The transition likely happened in the mid-to-late Archean[21–24] but the style of the earliest Archean tectonics may have been very different from that of today. It could have been on-off, with limited subduction surrounding big mantle swells[24] or dominated by "sagduction" or vertical lithospheric delamination[22] due to the predominance of plume-related activity[25,26]. In such regimes, progressive accretion of basaltic lavas on the surface and delamination of the deep lithosphere results in a limited amount of recycling of the surface-altered material (e.g., with respect to $\delta^{18}O$ and other proxies) into the convecting mantle. Shallow and flat subduction or plate stacking would also result in limited exchange[23–26].

With documented evidence of accretion of the first super-continent, Kenorland by 2.5 Ga[27] comes evidence of large lateral continental migrations and continental collisions, and modern-style subduction under the continents. Subduction is capable of accreting sediments and efficient delivery of surface-altered materials via melts and/or fluids, with both low- and high-$\delta^{18}O$, into the mantle and directly under the continental lithosphere. It follows, therefore, that the early Archean sub-continental mantle could have been less affected by subducting slabs, i.e., be more similar to lunar rocks, thus more closely reflecting the BSE. Finally, the earliest CLM (at least pre-3.8 Ga) may not be preserved because the Earth's earliest lithosphere went through a major resurfacing process in Eoarchean as indicated by Hf isotopes in zircons[9,11,24].

Of independent and particular importance is the question of potential differences between the Earth's earliest mantle, represented by xenoliths (fragments) of Archean age in volcanic rocks, and younger mantle samples. As the Archean mantle was 100–200 °C hotter than today[2,24,28], degrees of shallow mantle melting were greater (up to 40%) leaving behind extremely melt-depleted harzburgites[20,29,30]. As these highly refractory residues were Fe-depleted, they were less dense and accumulated to form the rigid and stagnant keels of buoyant Archean continental lithosphere[30]. This difference can be seen, for example, in the distinct composition (e.g., olivine Mg#, defined as Mg/(Mg+Fe)) of Archean and modern peridotites[20,29]. Formation and subsequent growth of the CLM and continental keels involve imbrication upon collisions of depleted and buoyant subducted oceanic lithosphere (including melt depletion directly at subduction settings followed by slab delamination) and thickening by compression around existing continental margins or subarc settings[28,31,32]. It can thus be expected that CLM rocks, like Earth's surface rocks, have different ages and details of chemical and isotopic composition related to their age and mode of origin. However, only recently have we started paying attention to this heterogeneity with respect to many elements and isotopic systems. Oxygen, the most abundant element on Earth should in theory reflect these changes.

Nonetheless, for the past ~30 years, the $\delta^{18}O$ values of peridotite xenoliths from many tectonic settings have been inferred to be roughly uniform (within analytical error ~±0.1‰) irrespective of age, tectonic setting, bulk composition, and mineralogy (e.g., spinel vs garnet facies, fertile vs. refractory) ever since new laser

fluorination methods to generate an expansive dataset of global peridotites were first employed[13]. Later studies of even strongly metasomatized, hydrous (amphibole-bearing) peridotites reported values within the canonical mantle ranges[33]. Previous studies, including those employing secondary ionization mass spectrometry (SIMS) (within ≥0.2‰ precision), found a range from 5.0 to 5.5‰ for olivine from different peridotite xenoliths, attributed to a subtle diversity of depletion/enrichment processes, but detected no covariation of $\delta^{18}O$ with modal or chemical composition, settings, or age[34,35].

Laser fluorination, with its error on the order of 0.01–0.08‰, continues to be the most precise method to resolve small 0.1–0.3‰ differences in $\delta^{18}O$. The analytical effort and a dataset presented here for 104 individual mantle xenoliths span 13 years but are based on a single set of standards, analytical methodologies, and normalization procedures (Supplementary Table 1) in a single lab of the University of Oregon thus avoiding inter-laboratory standardization/analytical $\delta^{18}O$ corrections that are important to recognize small trends. We report data for all samples that came to the lab without selection bias. These are well-characterized xenoliths with available modal, petrographic, and chemical data pertinent to the depletion/enrichment events (Mg#, $Al_2O_3$, CaO) as well as equilibration temperatures[36–41], and references in the supplementary). These samples are not uniformly distributed geographically, because of explosive volcanic eruptions: kimberlites and alkali basalts that can fragment the CLM and can carry mantle xenoliths to the surface require volatile-rich magmas. These, however, span many tectonic settings and regions on Earth, and occur above hot spots, at continental plate boundaries, and rift zones where explosive eruptions rapidly carry pieces of the mantle to the surface at rates of meters per second (e.g. ref. [42]). This rapid transport prevents extensive chemical and isotopic exchange between the host melt and the xenoliths, especially for major elements including oxygen. Finally, CLM fragments are much easier to find in and extract from, loose pyroclastic facies of volcanic eruptions than in massive (and slower cooled) lavas. This is why much work on mantle xenoliths is done on the best available samples at each locality worldwide and in specific regions. In such cases, the work is typically performed on the largest, freshest, or least altered mantle xenoliths from several such sites. Comparison with smaller, partially altered, but more abundant xenoliths from broader regions usually indicate that the selected best samples are representative of larger areas in a specific lithospheric block and tectonic setting (refs. [34,36–42] and references therein). This statement characterizes the suite of xenoliths we present here.

## Results

**Temporal $\delta^{18}O$ trends for continental mantle peridotites.** Figures 1 and 2 present temporal $\delta^{18}O$ trends for our worldwide peridotite xenolith suite from cratonic, off-craton, and mantle wedge settings spanning ~3 Ga. The formation (melt extraction) ages for cratonic xenoliths were estimated from Re-Os isotope data[36–41] (Supplementary Table 1) using Re-depletion ($T_{RD}$) estimates that presume complete Re extraction from residues of BSE melting and thus are only valid for highly refractory rocks (harzburgites and dunites). The $T_{RD}$ values are slightly lower than ideally more correct model Re-Os ages ($T_{MA}$) yet allow to account for common post-melting Re-enrichments in xenoliths due to metasomatism and host magma infiltration responsible for common aberrant $T_{MA}$ values[30,36,40]. For consistency, we attribute the same age to coeval rock types from each locality, e.g., 2.9 Ga and 2.6 Ga for dunites vs. 2.0 Ga for harzburgites from Udachnaya kimberlite pipe in the Siberian craton. These estimates match the main stages of cratonic crust formation supporting their validity[38–40].

Re-Os ages cannot be systematically used to date off-craton peridotites that range broadly in composition, but are generally more fertile than in cratons, hence valid $T_{RD}$ ages cannot be obtained. We presume that the off-craton CLM formed during the most recent crustal growth or accretion event in a given lithospheric domain[43]. Harzburgite xenoliths from the west Pacific represent residual oceanic mantle, likely melted first at mid-ocean ridges, then in the mantle wedge. The Kamchatka samples are presumed to be ~0.1–0.3 Ga old (the oldest large-scale oceanic crust in the region)[44]. $T_{RD}$ estimates (0.1–1.1 Ga) are used for the West Bismarck samples because some of them may come from older microcontinents[45].

When plotted vs age (Fig. 1) $\delta^{18}O$ values of olivine exhibit a temporal ~0.2‰ gradual decrease from 5.38‰ in the Mesoarchean to 5.21‰ in the Paleoproterozoic and 5.16‰ in the Phanerozoic. The $\delta^{18}O_{Opx}$ decreases as well. The bulk $\delta^{18}O$ values (estimated via modal abundances of minerals and their measured or computed $\delta^{18}O$ values at last equilibration temperatures) show a sub-parallel trend of comparable magnitude. When the dataset is organized by locality, to avoid overrepresentation of localities with more analyses, this leaves only 15 individual mantle domains sampled by each kimberlite or basalt (Fig. 2). The temporal negative trend persists with better $R^2$.

Importantly, the differences between the Archean and Proterozoic olivine are observed not only globally but locally in two kimberlite pipes in Siberia (Udachnaya and Obnazhennaya) (Fig. 1a)[37,40]. Olivine modal abundance in our predominantly harzburgitic dataset stays nearly constant at 76 ± 6% from 3 Ga to 2 Ga. Mg# in our dataset displays an overall decrease from 0.93 to 0.94 in the Archean-early Proterozoic to ≤0.92 after that, consistent with greater melting degrees and pressures for the ancient mantle[20,29]. The last equilibration temperatures estimated with the Ca-in-Opx thermometer are lower for the higher-$\delta^{18}O$ Archean cratonic xenoliths than for off-craton ones but display no trend vs. age (Supplementary Table 1). Different temperatures of storage do not have any $\delta^{18}O$ effect on the bulk, but colder peridotites would have lower $\delta^{18}O_{Olivine}$ and higher $\delta^{18}O_{Px}$ values (greater $\Delta^{18}O_{Opx-Ol}$, Fig. 3). The lower temperature should result in lower (not higher) $\delta^{18}O_{Olivine}$ values of olivine at identical $\delta^{18}O_{Bulk}$ (cf. Fig. 3) and normalization to a single temperature would increase $\delta^{18}O_{Ol}$ for the Archean olivine commonly stored in colder conditions of thicker continental roots, and further decrease it for hotter recent samples, amplifying the temporal trend for olivine.

To explore the validity of the discovered trends further, we performed thermodynamic calculations of the effects of melt depletion during mantle melting and its effects on residual harzburgite assemblage (Fig. 3).

**Mantle melting and $\delta^{18}O$ value of the starting peridotite.** Performed modeling of bulk, fractional, and fluxed melting of original lherzolitic peridotite reveals small associated oxygen isotopic effects on coexisting minerals, melt, and remaining residue bulk solid (Fig. 3). As the basaltic melt is 0.4–0.7‰ higher in $\delta^{18}O$ than the bulk[15], due to the predominant removal of higher $\delta^{18}O$ pyroxenes upon basalt production during peridotite melting, fractional melting with the removal of 30% of melt leads to a 0.12‰ decrease in the remaining bulk harzburgitic residue $\delta^{18}O$ values. The 45% melt removal and production of dunitic residue, carries its 0.19‰ $\delta^{18}O$ depletion. In addition, the noticeable and counter-intuitive feature of the peridotite melting process that has not been discussed previously to our knowledge is that the $\delta^{18}O_{Olivine}$ value increases during the formation of these melting residues (harzburgite and dunites), as $\delta^{18}O_{Olivine}$

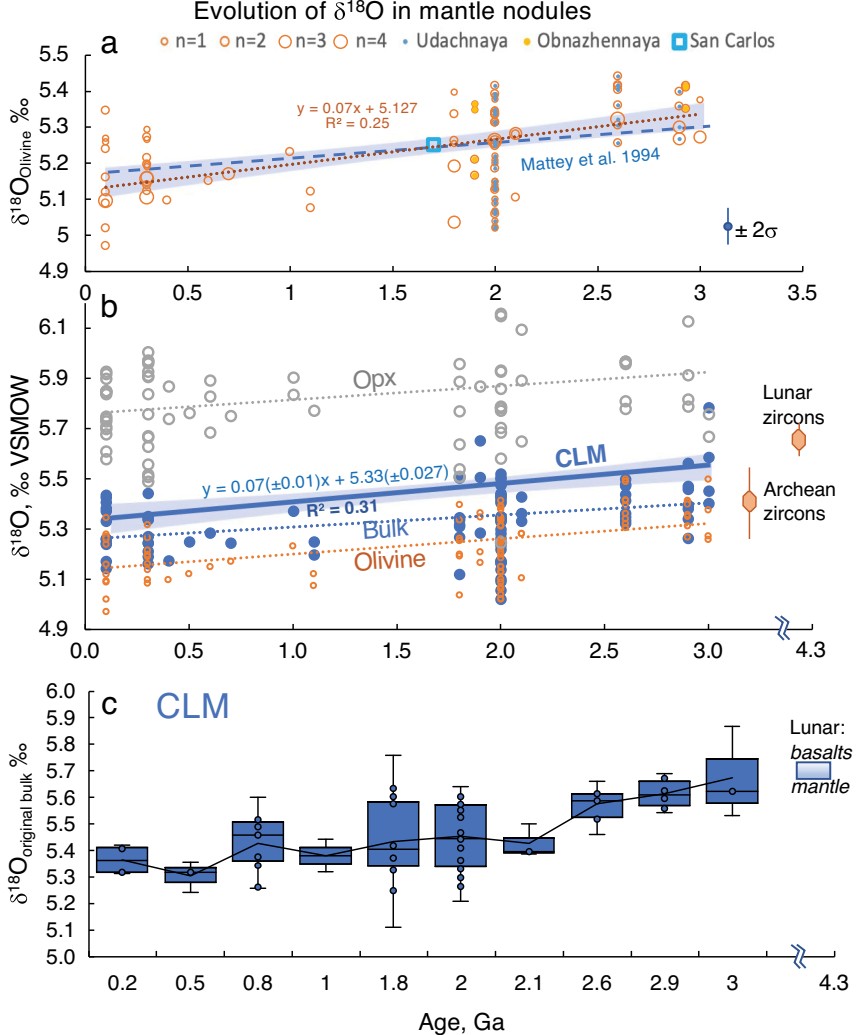

**Fig. 1 Evolution of oxygen isotopes in lithospheric continental mantle (CLM).** Temporal trends of $\delta^{18}O$ olivine (**a**, **b**), orthopyroxene and bulk (**b**), and CLM corrected for melt depletion (**b**, **c**) in studied mantle nodules. Bulk is computed based on modal mineral abundances and $\delta^{18}O$ values of olivine and orthopyroxene (T Supplementary Table 1). Zircon from the Archean Kaapvaal cratonic mantle is from ref. [50], and zircon and basalts from the moon are from refs. [12,51], lunar mantle estimate is based on zircon, olivine, and basalts (see the text). Notice decreasing trends (95% conf. interval error envelope in line fit) and inbox diagrams.

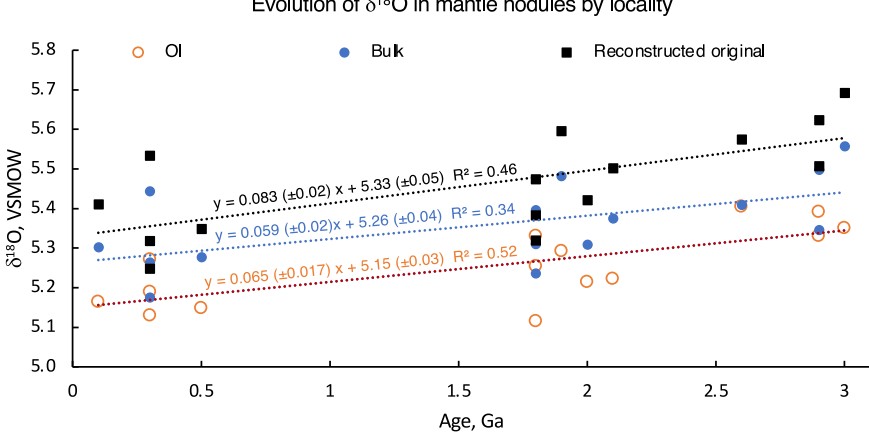

**Fig. 2 Locality-averaged $\delta^{18}O$ values for the studied samples of olivine (Ol), bulk, and reconstructed original undepleted peridotite (see Fig. 3).** Line fit statistics indicate decreasing trends.

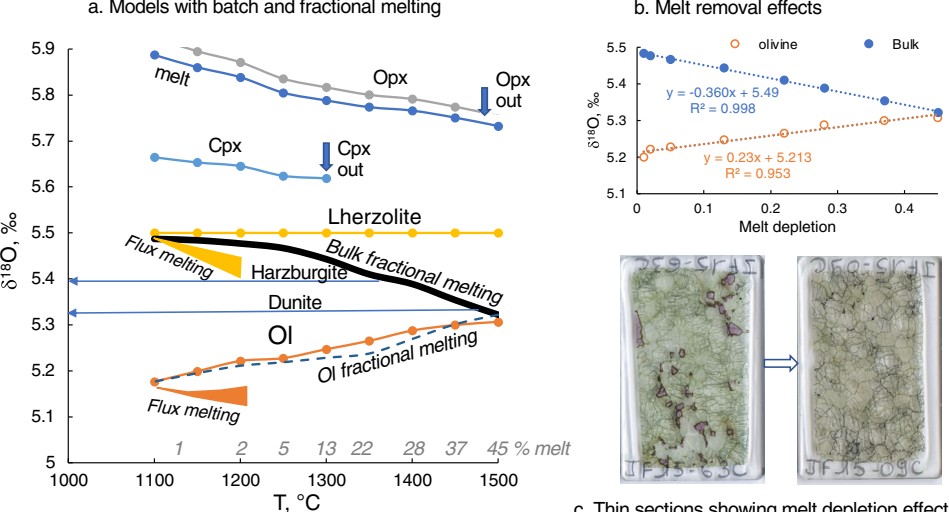

**Fig. 3 Computed isotope effects of batch (continuous lines) and fractional (dashed lines) peridotite melting from 0 to 45‰ on the equilibrium isotope values of melt and the residual mineral assemblage. a** Isotope effects on minerals and the residual bulk. Starting composition at low-T has 56% olivine, 26% Opx, 15% Cpx, and 3% of Spinel. Isotope fractionation factors are from refs. [62,63]. For melt, we assumed A = 1.3. Isotope effects are computed against phase change and proportions (% melt) trends observed in ref. [64] in their peridotite melting experiments, also modeled in ref. [65] in pMELTS. Fractional melting assumed the removal of equilibrium fractional melt shown on the diagram. Flux melting trends are computed using Rayleigh melting by water fluxing and up to 5% fractional melt removal with shown effect on bulk peridotite and residual olivine $\delta^{18}O$ values. In this case, it is assumed that water fluxing generates silica-rich andesitic hydrous melt/fluid with $\delta^{18}O = 6.1–7.3‰$ in equilibrium with the ambient assemblage. The upper bound of the flux melting triangle is computed by assuming decreasing $\delta^{18}O$ from 7.3‰ to 6.1‰, while the lower bound assumes that the hydrous melts maintain a constant high $\delta^{18}O = 7.3‰$. Notice that melt depletion results in lowering bulk $\delta^{18}O$, but it also results in higher modal olivine and the $\delta^{18}O_{Ol}$ values which are higher (closer to the bulk), while $\delta^{18}O_{Opx}$ is also higher; $\Delta^{18}O_{Opx-Ol}$ are constant. **b** effect of melt depletion along the melting path shown in (**a**) for $\delta^{18}O_{Ol}$ or $\delta^{18}O_{bulk}$. Notice the nearly linear change. Upon melt extraction, $\delta^{18}O_{dunite} < \delta^{18}O_{harzburgite} < \delta^{18}O_{lherzolite}$. cooling will not affect $\delta^{18}O_{bulk}$ (blue horizontal lines) but will increase $\Delta^{18}O_{mineral-mineral}$. fractionation as is shown. **c** Thin sections of samples showing enriched and depleted harzburgites that lost melt, and the final residual assemblage is olivine-richer. The horizontal width is 3 cm.

becomes closer to the bulk values while its modal proportion increases. This simple behavior of oxygen isotope repartitioning during peridotite melting accounts for small 0.05–0.1‰ corrections due to the melt removal effects allowing reconstruction of the original peridotitic bulk before melting.

This estimation of approximate melt depletion is given in Supplementary Table 1 and is based on $Al_2O_3$, MgO, and Mg# by using the experimental data of ref. [29]. It leads to ~45% melt extraction estimates for dunites and lowest-Al (<0.5) harzburgites, 40% for low-Al, high-Mg# (>0.925) harzburgites, 30–40% for Proterozoic harzburgites with 0.5–1% $Al_2O_3$, 20–25% for moderate-Al harzburgites, 30% to all Cenozoic Avacha harzburgites (28% to those with 0.8% $Al_2O_3$). Although estimates of the melting degrees are likely accurate to ±5–7%, we consider them sufficiently robust because associated effects on $\Delta^{18}O_{depleted-original}$, and on $\delta^{18}O_{Olivine}$ or $\delta^{18}O_{bulk}$ are simple and small (Fig. 3b). The 25% prior melt removal leads to a 0.05–0.06‰ increase in $\delta^{18}O_{Olivine}$ and 0.09‰ decrease in $\delta^{18}O_{bulk}$ peridotite values. The absolute majority of continental peridotites including those studied in this work are represented by melt-depleted harzburgites with a 76% modal olivine and with broadly comparable 28–40% estimated melt depletion[29], so a total change between $\delta^{18}O_{depleted}$ and $\delta^{18}O_{original}$ peridotite of different groups is −0.05 to −0.06‰ across the entire database.

The Archean samples come as ~10% more melt-depleted than post-Archean (Fig. 1c and Supplementary Table 1), perhaps due to the hotter regime in the Archean leading to a greater degree of mantle melting and komatiitic melt extraction (e.g., ref. [24]). Applying this melt depletion correction leads to 0.03–0.05‰ additionally higher $\delta^{18}O_{original}$ Archean peridotite, and the overall estimate of temporal $\delta^{18}O_{bulk}$ peridotite difference between the Archean and Phanerozoic samples is further increased to

0.18–0.2‰ (Fig. 1). The t test of comparing Archean vs post-Archean returns P values of <0.001 (Supplementary Table 5). The earliest Archean xenoliths thus return $\delta^{18}O$ values comparable to that of the Moon and are used here to estimate the BSE (Fig. 1). Melt correction leads to a more pronounced $\delta^{18}O_{original}$ negative trend with age. Therefore, this modeling presented in Fig. 3 precludes an interpretation of the discovered temporal trends on Figs. 1 and 2 due to the effects of melting.

**Comparison with other datasets.** The decreasing olivine $\delta^{18}O$ trend that we observed made us search for similar trends in other published datasets, of which ref. [13] is the most comprehensive and done in the same lab. When assigning ages to their xenolith suites, based on studies of similar xenoliths from the same or analogous age and tectonic provinces (Supplementary Table 3), which has not been done before, we observe a trend similar to ours (Fig. 1 and Supplementary Fig. 2): decreasing $\delta^{18}O_{Olivine}$ and $\delta^{18}O_{Bulk}$ values and constant or even decreasing $\delta^{18}O_{Opx}$ values.

We experimented with an alternative age assignment to the samples studied here and to the dataset in ref. [13], by simply using model formation ages of the host lithosphere (e.g., ref. [46] and Supplementary Figs. 1 and 2). This results in similarly decreasing trends, signifying the conclusion that the obtained: subtle temporal trends are not fortuitous results of age assignments.

The average for $\delta^{18}O_{Olivine}$ in our entire datasets (5.224 ± 0.120, s.d., n = 174) is heavier by 0.04‰ than that for data presented in ref. [13], which we attribute to different sample suites and small differences in standardization. The computed average bulk values of 5.34 ± 0.13‰ are identical to ref. [13]. We thus consider the average values of the two datasets as compatible and provide the best current estimates of the $\delta^{18}O$ in the average lithospheric

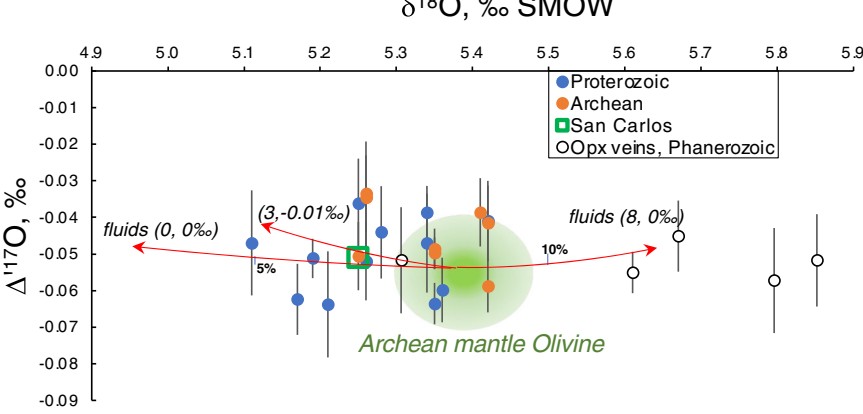

**Fig. 4 Triple oxygen isotope analyses of Archean and post-Archean olivines and orthopyroxenes showing overlap with San Carlos olivine.** Expected trends to higher-$\Delta'^{17}O$ and lower and higher $\delta^{18}O$ subduction fluids, derived from subducting slabs are shown. The $\delta^{18}O$ in orthopyroxenes is higher than equilibrium olivine by 0.5–0.7‰ and nearly identical in $\Delta'^{17}O$ at mantle temperatures (Fig. 3, ref. [66]).

mantle (Supplementary Table 4). Below we further consider its evolution.

The 5.37‰ value for Archean olivine and 5.44‰ bulk value for mean Archean refractory peridotite of our dataset with the average 79% modal olivine and 16% modal orthopyroxene enables correction for a decrease in $\delta^{18}O$ due to prior melt depletion as is explained in Fig. 3a, b. Translating this harzburgitic bulk into a typical original lherzolitic assemblage of 56% olivine and 26% orthopyroxene[30] leads to a 0.12‰ increase in the value of the original lherzolitic bulk peridotite. We thus obtain a value of 5.57‰ as the best estimate of the primary mantle source (Fig. 1c) that characterizes the original BSE prior to melting in the Eorchean. The olivine $\delta^{18}O$ value of 5.37 ± 0.07‰ (1 s.d., $n = 20$) recorded in the Archean olivine would stay the same and may thus reflect the best estimate for olivine in the BSE.

**Triple oxygen isotopes**. Triple oxygen isotope analyses of a selection of olivines (Fig. 4 and Supplementary Table 2) yield an overlap in values: $\Delta'^{17}O_{0.5305} = -0.047 \pm 0.011$‰ ($n = 9$) for Archean vs $-0.051 \pm 0.010$‰ ($n = 12$) for the early-Paleoproterozoic samples. The value for San Carlos Olivine run as standard, likely sampling younger off-craton Phanerozoic (≤1.8 Ga) mantle of N America[46], measured in the University of Oregon and two other labs are $-0.051$‰[47]. Vein orthopyroxenes with minor amphibole from the Kamchatka arc, most likely formed due to subduction fluid/melt percolation[48], yield a mantle-like value $-0.052 \pm 0.005$‰. Modeling demonstrates that fluxing with fluids with any reasonable $\delta^{18}O$ (0 to +10‰) or $\Delta'^{17}O$ (+0.02 to $-0.02$‰) values will have a far greater impact on $\delta^{18}O$ than on $\Delta'^{17}O$, resulting in largely constant mantle-like $\Delta'^{17}O$. Our data confirm the recent result in ref. [49] on limited $\Delta'^{17}O$ mantle ranges.

**Earth and Moon initial $\delta^{18}O$ values: BSE = BSM**. Significant attention has been recently paid to recognizing small $\Delta'^{17}O$ differences between the Earth and the moon, pertinent to the original $\Delta'^{17}O$ composition of Thea and the degree of homogenization of magma ocean, as well as the late-veneer $\Delta'^{17}O$ composition[3–6,49]. However, the $\delta^{18}O$ value of the earth's mantle and its predominant rock on earth, peridotite has not been revisited since 1994[13]. This work fills this gap in an attempt to reconstruct the original BSE oxygen isotopic values, and we show here that it is higher than commonly assumed and is closer to that of the moon at 5.57‰ (Fig. 2).

Constraining $\delta^{18}O$ for mantle older than 3.0–3.5 Ga is hampered by the lack of dated peridotites and unaltered olivine, formed at the time. Here, we use crustal zircon as a proxy for mantle olivine, since zircons of such age exist in mantle-derived crustal rocks, survive alteration, and because zircon-olivine O isotope fractionation at mantle temperatures is very small and is well-constrained (0.1 ± 0.1‰[50], see below). For example, Kaapvaal craton zircons of the ~3.2 Ga Archean age exhibit the range of 5.32 ± 0.17‰[12]. Lunar zircon of presumably 4.4–4.3 Ga age has $\delta^{18}O$ of 5.61‰[12]; these values are 0.3‰ heavier than typical terrestrial mantle zircons (5.3‰[50,51]), most of which are Phanerozoic. The $\delta^{18}O$ composition of typical 4.3–3.8 Ga lunar basalts and zircons can be used to estimate the $\delta^{18}O$ value of the lunar (and coeval terrestrial) mantle by a simple isotopic mass balance approach involving computing the CIPW mineral norm of basalts, taking a weighted average of $\Delta^{18}O_{\text{CIPW mineral – zircon fractionations}}$ using published fractionation factors ([14,15,50] at assumed temperatures, see Fig. 3). Such computation results in $\Delta^{18}O_{\text{lunar basalt-peridotite}}$ of 0.02 to 0.12‰ at 1400–1450 °C by targeting common lunar basaltic magma types with lower values being characteristic for high-Ti and higher for low-Ti basalts. As lunar zircons are lighter by 0.08–0.09‰ than lunar basalts[12], the $\Delta^{18}O_{\text{lunar zircon-peridotite}}$ is 0.01–0.02‰, indicating essentially that $\delta^{18}O_{\text{lunar zircon}}$ is identical to $\delta^{18}O_{\text{peridotite}}$. Therefore, the lunar zircon plotted in Fig. 1 serves as a very good proxy for the lunar (and perhaps terrestrial) mantle at 4.3–3.8 Ga.

Another way to resolve these small but important differences in lunar and terrestrial peridotites is to directly consider terrestrial and lunar basalts derived from their respective mantles. Typical 4.3–3.8 Ga low-Ti lunar basalts are 5.69‰, 0.12‰ heavier than the mode of modern terrestrial MORB (5.57‰, ref. [12]). There are no coeval and unaltered 4.3–3.8 Ga terrestrial basalts to precisely compare with lunar basalts, but lunar basalts can be used to suggest that coeval terrestrial basalts of an equivalent degree o mantle melting of the Hadean age, were also higher in $\delta^{18}O$.

The higher $\delta^{18}O$ values for lunar zircons and basalts directly support the conclusion of our present study that mantle peridotites (parental to basalts) evolved temporally to lower $\delta^{18}O$ values, and thus the signature of basalts is directly inherited from peridotites. Both planets had magma oceans after the Giant impact of 4.44 Ga and thus likely had initially similar upper mantles, and basaltic upper crusts with respect to major element oxygen[5]. Triple oxygen isotopic effort to find differences between the earth and the moon resulted in almost complete overlap[6].

As the Moon has not had plate tectonics, it thus provides a benchmark for the pre-plate tectonics on Earth since both planets

re-accreted after the Giant Impact (4.44 Ga) and had contemporaneous magma oceans[3,4]. These heavier values agree with our extrapolated estimates of $\delta^{18}O_{BSE} = 5.57‰ \pm 0.07‰$ (based on peridotites with propagated errors) very well.

Our estimate of the $\delta^{18}O_{BSE}$ is of course related to the timing of initiation of plate tectonics, which is a matter of significant debate[7,11,22–26]. If significant subduction started much earlier than 3 Ga, and TRD ages were also reset, then the pre-plate tectonic $\delta^{18}O_{CLM}$ would have been slightly higher than we estimate here. For example, extrapolation of the linear trend for CLM in Fig. 1 would result in 5.61‰ at 3.8 Ga and 5.63‰ at 4.1 Ga, but within error of our measured Archean estimate of 5.57 ± 0.07‰. We notice, however, that the temporal change in $\delta^{18}O$ in the Archean samples from 3.0 to 2.6 Ga is less steep than the change between the Archean and Proterozoic and Archean vs all post-Archean samples (Fig. 1c), and so extrapolation $\delta^{18}O$ value is less than the above estimates. Slab dehydration on early hotter Earth would likely happen under shallower conditions (24) and the oxygen isotope effects that we describe below, were likely less prominent. Perhaps more importantly, the oldest peridotite samples in our collection are ~3 Ga and worldwide there are no CLM peridotite xenoliths with robust bulk-rock Re-Os ages significantly more than 3 Ga[30,41]. This perhaps reflects that there may not be much left of surviving earliest Eoarchean and Hadean CLM as it was recycled back into the convecting mantle immediately after the Hadean (9,11, 24), annihilating the effects of early subduction (if any) on crust-CLM oxygen isotope repartition. Thus, if the plate tectonic started at 3.8–4.1 Ga (the earliest suggested estimates[7,24]) the $\delta^{18}O$ of the CLM was not significantly changed.

**Origin of decreasing $\delta^{18}O$ trend in mantle peridotites.** It should be noted that most of the samples in the peridotite collection brought up by basalts and kimberlites in this work represent not the entire Earth's convecting mantle (difficult to sample through time) but only its more or less melted residues of the Earth's upper boundary layer of different age, which form 100–200-km-thick continental lithospheric mantle. The decreasing $\delta^{18}O$ trend that we first observed by assigning age to studied collections (Figs. 1 and 2 and Supplementary Figs. 1 and 2) is not due to melting, cooling, or variable effects related to melt depletion (Fig. 3). We demonstrated that accounting for greater melt depletion in the Archean only enhances the temporal trend to 0.2‰. The only terrestrial reservoir that is lower in $\delta^{18}O$ than the mantle is the terrestrial hydrosphere, and thus the most parsimonious interpretation of the trend is that of putatively accumulated effects of water–rock interaction.

Thus, we next consider the possible mechanisms for an external low-$\delta^{18}O$ hydrospheric fluid interaction with the peridotites and their time-integrated effects, as well as recycling of hydrothermally altered lithospheric materials (Fig. 5) and describe likely processes that were in operation.

First, direct hydrothermal alteration (serpentinization and chloritization) of the shallow mantle in areas accessible to seawater percolation occurs today, in Cenozoic ophiolites (ref. 14 and in the geologic history at mid-ocean ridges, as well as, importantly, in areas where plates bend in front of subduction trenches[24,52]. Any such hydrothermal interaction at $T > 200\,°C$, plausible in terms of typical sub-Moho temperature gradients, would generate lower-$\delta^{18}O$ peridotites[14,15]. Subsequent deserpentiziation reaction and restoration of the olivine and orthopyroxene would preserve the original low-$\delta^{18}O$ birthmark[49]. These low-$\delta^{18}O$ peridotites could have accreted under the continents upon initiation of horizontal motion.

Second, such low-$\delta^{18}O$ portions of subducted slabs expel low-$\delta^{18}O$ fluids and thus modify the $\delta^{18}O$ of the overlying mantle

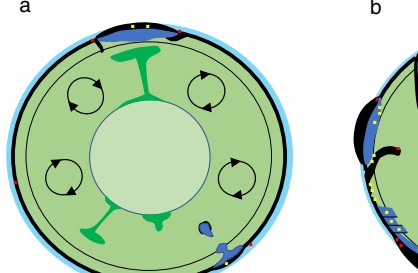
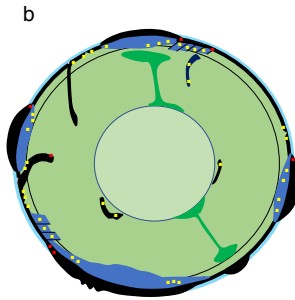

**Fig. 5 A cartoon that explains the cooling of the Earth and rehydration of the mantle by low-$\delta^{18}O$ fluids in spreading centers[14,15], plate bending zones[52], and subduction zones[53,56], generating progressively lower $\delta^{18}O$ (and overall heterogeneous) continental lithospheric mantle (darker blue) after initiation of plate tectonics.** $\Delta'^{17}O$ is not significantly modified by these processes (Fig. 4). Color code: yellow: low-$\delta^{18}O$ peridotites, red: high-$\delta^{18}O$ sediments, basalts, and eclogites, green: primitive mantle plumes, light blue: oceans. **a** Early Earth and moon regimes: degassed mantle, plume tectonics, rudimentary subduction around plumes, and intense hot mantle convection. The $\delta^{18}O$ of the original peridotites is 5.57 ± 0.07‰, and Bulk Silicate Earth=Bulk Silicate Moon. **b** Modern plate tectonics regime: rehydration of mantle, plate accretion, and imbrication, lower-$\delta^{18}O$ peridotites in the subcontinental lithospheric mantle samples by studied xenolith suites.

wedge peridotites. Given the $\delta^{18}O$ distribution in the altered oceanic crust and sediments on it, which ranges from high-$\delta^{18}O$ at the top to low-$\delta^{18}O$ in the chloritized and serpentinized interior[14,16,53], upon subduction and slab heating the high-$\delta^{18}O$ fluid will be lost first at the shallow fore-arc, while dehydration of the low-$\delta^{18}O$ interior of the slab would occur deeper into the peridotitic mantle wedge as the slab geotherm progressively increases[54]. The modern analogs to Archean subduction are modern "hot" subduction zones, such as the Central American Arc[53]. Eiler et al.[53] argued that the subtly low-$\delta^{18}O$ value of the magmas in the central part of this arc results from low-$\delta^{18}O$ fluids, and greater (2–4%) addition of such fluids corresponded to greater (15–25%) degree of mantle melting. Our sample collection has two Phanerozoic arc-related xenolith suites: harzburgites from the Kamchatka[44] and the West Bismarck arcs[45]. Both occupy the low-$\delta^{18}O$ end of our sample array, supporting the proposed scenario of fluxing by low-$\delta^{18}O$ fluids. In other, shallower areas of the arc, fluids can be high-$\delta^{18}O$ and normal (mantle-like)-$\delta^{18}O$, contributing to higher $\delta^{18}O$ values of accretionary sediments and the crust. Besides these two localities studied in this work, strongly re-worked mantle peridotites elsewhere with deformed and modally metasomatized at high temperature and pressure textures, show shifts to lower (not higher) $\delta^{18}O$ values (down by 1‰[55]). The preentrainment metasomatism is invoked to occur by deep subduction fluids shortly before magmatic entrainment, which thus also appears low-$\delta^{18}O$.

**Discussion**

Here reported a temporal decrease in $\delta^{18}O$ value of peridotites, dominated by continental harzburgite, is likely due to a billion-year record of seawater-peridotite interaction in areas of spreading and plate bending, and the subduction-accretion of these materials to the CLM (Fig. 5). If we consider the entire 100 km-thick section of such mantle, at present subduction length of 62 K km and rates of 8 cm/yr it would take 2 Gyr to lower the value of peridotite by 0.15‰. This computation assumes that the remaining 1–2 wt% of $H_2O$ is released from the upper 10 km of the slab into the overlying peridotite as a low-$\delta^{18}O$ fluid (0 to

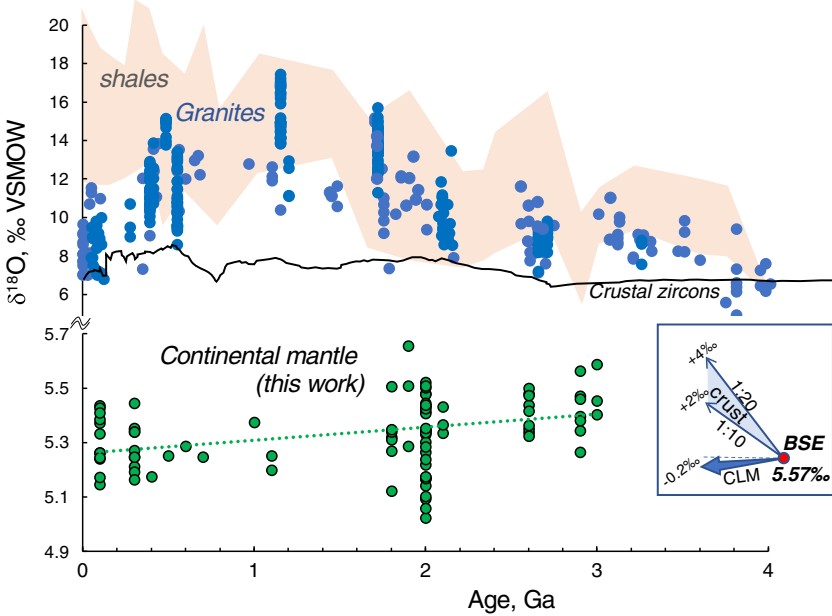

**Fig. 6 Co-evolution of the continental lithospheric mantle (CLM, this work) and the Earth's crust as expressed by three proxies: granites, bulk shales[57], and zircons[67].** Note change in $\delta^{18}O$ scale. Inset shows mass balance of CLM and crust during growth and evolution; 1:10 and 1:20 are assumed crust/CLM proportions based on the modern average lithospheric thickness of 150 km. BSE is estimated in this work and corresponds to the starting point of crustal growth after the Giant Impact. Unidirectional temporal loss of low-$\delta^{18}O$ to the large reservoir of CLM helps to explain the accumulation of heavy $\delta^{18}O$ in the crust and potentially hydrosphere (e.g., 57–60).

+2‰) due to the breakdown of serpentine and chlorite ~100–200 km below the mantle wedge[54,56]. This estimate is likely robust within a factor of 2–3 depending on the choice of water content, its isotopic composition, and the total volume of the mantle affected. Perhaps more importantly, only a certain fraction of the mantle under volcanic fronts may be affected, compared to neighboring areas, or by different fluids. Besides subduction, as is mentioned above, direct seawater-peridotite interaction at spreading and plate bending zones, followed by plate stacking and imbrication would potentially produce a range of $\delta^{18}O$ for xenoliths from a single locality. This depends on the mechanisms and timescales of fluid migration/fluid-rock reaction and subsequent recrystallization of fluxed peridotites. This explains resolvable $\delta^{18}O$ heterogeneity observed among xenoliths from a single kimberlite pipe (Udachnaya and Obnazhennaya (Fig. 1a[38–40]) or xenoliths from variously fluxed mantle wedges in the Phanerozoic Kamchatka or West Bismarck arcs[44–46]. This is an important observation that can also explain moderate degrees of $\delta^{18}O$ diversity in basaltic melts of a single locality that we call "natural $\delta^{18}O$ noise" generated from a moderately laterally diverse $\delta^{18}O$ shallow (continental mantle-derived) mantle source.

The trend established here for the isotopic composition of a major element, oxygen, supports a temporal oxygen isotopic separation between the upper mantle and the crust via gradual cumulative and putative effects of accreting low-density, high-$\delta^{18}O$ sediments, and upper basaltic portions of the subducting slabs at shallow crustal levels, and supplying low-$\delta^{18}O$ peridotitic material and fluids into the CLM. We here suggested that the described process of $^{18}O$ separation may also explain the contemporaneous increase of $\delta^{18}O$ in the continental crust (Fig. 6) and the remaining hydrosphere, as documented by surface sedimentary archives[57] and refs therein], a long-standing controversy in Earth sciences[58,59]. Although not considered in this work, sunken slabs with residual water and predominantly low-$\delta^{18}O$ values would additionally transfer more $^{18}O$-depleted oxygen into the predominant volume of the convecting mantle (Fig. 5), working in the same direction of contributing to the overall

increase in $\delta^{18}O$ of the crust, hydrosphere, and surface materials[57,58]. Described low-$\delta^{18}O$ unidirectional flux from the surface to the shallow and deeper mantle is likely firmly linked to the intensity of plate tectonics. As it mostly comes with water, low-$\delta^{18}O$ flux is correlative to mantle rehydration, which was suggested earlier to explain the secular drop in the sea level[60] contributing to continental emergence after the Archean[61]. Given that the mass of the CLM is 10–20× that of the continental crust, a decrease by ~0.2‰ in $\delta^{18}O$ of the CLM on a Ga time scale that we documented here, would suggest an increase of up to 4‰ in the crust (Fig. 6) by a simple crust-CLM mass balance, without taking into account unidirectional loss of the subduction-affected materials (both low and high-$\delta^{18}O$, hydrous and anhydrous, e.g., Fig. 5) to the deeper convecting mantle. Such a calculation assumes that most of the oxygen isotope mass balance proceeds via surface tectonics and fluid transfer (hydration–rehydration) and is mostly contained within lithospheric plates of 100–250 km. A similar approach for other chemical and isotopic systems[56] that record interaction with the hydrosphere may serve to test the above conclusion and better quantify its mechanisms as well as the onset of mantle fluxing and rehydration of the mantle after the beginning of modern-style plate tectonics.

## Methods

**Analytical methods for $\delta^{18}O$.** Oxygen isotope measurements for $\delta^{18}O$ relied on 1–1.5 mg of material, predominantly single crystals of olivine, and were performed by laser fluorination and gas-source mass spectrometry (MAT253) at the University of Oregon, and the effort spanned 13 years (Supplementary Table 1). Coexisting orthopyroxenes, clinopyroxenes, and garnets were also analyzed in many samples. We used purified $BrF_5$ as a reagent and boiling Hg diffusion pump to get rid of excess $F_2$ gas, then converted purified $O_2$ into $CO_2$ and run it in a dual-inlet mode on MAT253 mass spectrometer, integrated with the vacuum line, as this method is most precise for $\delta^{18}O$ determination[15,16]. Samples yields were measured in a calibrated volume using a Baratron gauge and were >90%, and when plotted vs $\delta^{18}O$ demonstrated no correlation. San Carlos Olivine ($\delta^{18}O = 5.25$‰, UWG2 garnet, 5.80‰[50], and an indoor UOG garnet standard ($\delta^{18}O = 6.52$‰) calibrated relative to the other two were used to calibrate the data on VSMOW scale. Each session included analyses of 4–6 standards, and correction for day-to-day variability was 0 to 0.2‰. Standard $CO_2$ gas was used as a working standard and it was periodically rerun against OZTECH $CO_2$ gas. Errors for standards in individual

sessions ranged from ±0.01 to ±0.11‰ and on average are ±0.06‰, 1 s.d. Samples from sessions with worse precision on standards were rerun and most data reported in Supplementary Table 1 represents duplicates with some triplicates/quintuplicates run in different sessions and are averaged with shown 2 s.e. on replicate measurements. Using $\delta^{18}O$ values for the same samples run on different sessions is a preferred way to recognize differences between samples.

**Triple oxygen isotopic methods.** For triple oxygen isotope analyses, generated gas was run as $O_2$ in sessions with SCO standard ($\Delta'^{17}O = -0.052$‰[47], and utilized a triple O continuous flow line constructed for the most precise triple O measurements[49,57] at the University of Oregon. The generated $O_2$ gas was put through the 8 ft long gas chromatographic column at room temperature for its purification from NFC compounds. Generated gases were additionally frozen on a 5 A zeolite and then released into the bellow of the mass spectrometer by LN2–ethanol mix over 10 min. GC and zeolite traps were degassed with flowing He at 200 °C for 15–20 min between each sample. The purified $O_2$ gas was run five times eight cycles in a dual-inlet mode, against a well-calibrated VSMOW University of Washington oxygen gas standard on a MAT253 isotope ratio mass spectrometer. Triple O isotope values for $\delta^{18}O$ and $\Delta'^{17}O$ are reported in Supplementary Table 2.

**Statistical handling of data.** Data handling included a statistical analysis that was performed by simple linear trends fitting (with error envelopes and line fit statistics) of the entire dataset and split (location-based) sub-datasets (Figs. 1 and 2). No data were excluded from the analyses, and we report all data accumulated over 13 years. We performed $T$ tests when comparing data of $\delta^{18}O$ and $\Delta'^{17}O$ for peridotites of different ages (Supplementary Tables 3 and 5) and used $P$ value as criteria to accept or reject the difference.

## Data availability

The Oxygen isotopic data generated in this study are provided in the Supplementary Information/Source Data file.

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

## Acknowledgements
We thank Bertrand Moine and Maya Kopylova for sample donation, Luc Doucet and Zhe Liu for assistance with sample processing, and Jim Palandri for analytical help, Chris Spencer is thanked for constructive comments. We thank Funding: US NSF EAR 1833420 (to I.B.), Chinese Academy of Sciences President's International Fellowship Initiative, Grant No. 2017VCA0009 (to D.I.).

## Author contributions
Conceptualization: I.N.B.; analytical research: I.N.B., D.A.I., and P.M.E.T.; samples and fieldwork: D.A.I. and A.G.; writing—original draft: I.N.B.; writing—review and editing: I.N.B., D.A.I., P.M.E.T., and A.G.

## Competing interests
The authors declare no competing interests.
