## [Peer Review File · Nature Communications]

Oxygen isotope ($\delta^{18}\text{O}$, $\Delta^{17}\text{O}$) insights into continental mantle evolution since the ArcheanREVIEWER COMMENTS

Reviewer #1 (Remarks to the Author):

Review NCOMMS-22-03547-T

Author: Ilya Bindeman

Fig. 1. With the exception of 1 data point at 2.1 Ga, a single straight line can be fit through the data, with no bump at 2.5, so it is hard to interpret this data as a sudden start of tectonics at 2.5 Ga, with none before.

Abstract

Written well, but please note the end of the Archean is not 2.5-2.0 Ga, it is 2.5 Ga.

The origin of the CLM is assumed to be accreted melting residues, but there are some other models and choices, and they are not all mutually exclusive. It is more complex.

So already there is a model assumption that the oldest CLM was not influenced by subduction. It should be discussed that in some views, subduction was operating since the Hadean. Maybe it was just less-influenced by shorter exposure to subduction?

BSE abbreviation is used before it is defined

Line 25. Yes, from looking at the data plots, it seems to be a general progression (nearly linear) with no real obvious jump at the Archean / Proterozoic boundary. From examining the data and figures first, before the text, I feel perhaps that the authors are using one of the popular models of the initiation of plate tectonics (at 2.5 Ga) and trying to make their data fit that model, but only considering the data from O isotopes. But, lets see. Topic is interesting.

Introduction

Line 50-55. Assuming, taking for granted that plate tectonics started in the Archean, or as here, at the Archean Proterozoic boundary, is a highly debated issue, and one of the most contentious in Earth Sciences. Only one reference is given, and it is one that is widely debated by the geological community, but perhaps it fits the model the authors are trying to support. At the very least the authors must discuss this issue, and bring up the various possibilities, and then test if their data supports any of the models. As it stands now, this is a fatal flaw in the manuscript, to assume this is correct, then take a gradual trend in the O data, and say it indicates plate tectonics started at 2.5 Ga. If that is so, the data must be consistent with all of the other evidence, geological, geochemical, and so on, much of which indicates a much earlier start to some form of subduction and plate tectonics. Please see recent reviews by Harrison 2020 (<https://dx.doi.org/10.1007/978-3-030-46687-9>), Keller and Harrison in PNAS (<https://doi.org/10.1073/pnas.2009431117>) and Windley in PR (doi.org/10.1016/j.precamres.2020.105980). If the starting point is not assuming that plate tectonics started at 2.5 Ga, then the authors could make a more scientific analysis, and might find something different, perhaps (as the data suggests to me) a slow gradual change in the O isotopes induced by a steady stream of subduction since the earliest records.

As for “sagduction”, that phrase is popular in some circles, but the process is poorly defined, and has never been observed or proven anywhere.

Why are the xenoliths of “tentative” Archean age?

65-70. As mentioned above, there are also several models for the formation of lithospheric keels, not just this simple one you list here. To be fair, you should discuss them, and discuss if your O isotopes could tell the difference. What about the evidence from Shirey and Richardson for subduction back to at least 3.1 Ga based on the isotopes of eclogitic and peridotitic inclusions? What about the models of keel formation by slab stacking?

I think this all needs to be reframed to test these models, not to assume one of the most controversial models is correct, then say your data supports it, when there is barely any change at the moment you say the planet suddenly changed from a non-plate tectonic world to a plate tectonic world with subduction recycling.

Results

The data, and the discussion of the data is all fine. So the data show a gradual decrease in O18 with time, at a fairly constant rate through the whole 3 billion years of the data sampling period.

For the Archean, you may do better to discuss the Neoproterozoic, Mesoproterozoic, and Paleoproterozoic, since the mantle temperature and melting degree surely changed a lot through those 1.5 billion years. By stating the change this way, as the change from Archean/Proterozoic, the authors are giving the readers the wrong impression that something suddenly happened. This is not consistent with the data shown, or with many other data sets not considered in this analysis.

Discussion

Line 170. There are peridotites of the age range of 3.0 – 3.5, and older. I am not sure why the authors say there is a lack of olivine formed at that time. Maybe say you have no data on samples from that age.

I am uncomfortable with the assumption here, of using the O from olivine from younger than 3.0-3.5 Ga, and using the O isotopes from zircon for 3.0-3.5 and older, to derive a single curve (line).

Summary

The data presented is of high quality, and shows a gradual trend over the course of 3 Ga. This could be related to continuous subduction over 3 Ga.

The authors have made too many assumptions and simply accept that plate tectonics started at the Archean/Proterozoic boundary, instead of using their data to test that hypothesis. If they reframed their argument as a test of this hypothesis, and linked better with other data sets, both geochemical and geological, I think they would have a stronger paper, and a different answer.

Reviewer #2 (Remarks to the Author):

Dear Editor,

Below is my review of Oxygen isotope ($\delta^{18}\text{O}$, $\delta^{17}\text{O}$) insights into continental mantle evolution since the Archean by Bindeman et al., submitted to Nature Communications. This manuscript provides convincing evidence for a decrease in the $\delta^{18}\text{O}$ composition of the continental lithospheric mantle as probed by

crustal xenoliths. Below are some minor suggestions on how this manuscript can be improved and some of their main points clarified. Although the paper is well written, there are a number of places where the verbiage is vague and imprecise. Nevertheless and notwithstanding my longwinded comments below, I see no reason why this paper cannot be published with minor revisions.

Title/Abstract: The title highlights the $\delta^{18}\text{O}$ and $\Delta^{17}\text{O}$, but the abstract doesn't discuss the $\Delta^{17}\text{O}$ at all. The $\Delta^{17}\text{O}$ shows minor variation in comparison to the $\delta^{18}\text{O}$. This is a significant finding in the light of previous $\Delta^{17}\text{O}$ work showing shifts in $\Delta^{17}\text{O}$ of shale and granite through time. The absence of any change in $\Delta^{17}\text{O}$, while the $\delta^{18}\text{O}$ does show a shift, should be highlighted in the abstract.

The paper focuses a lot on changes that occurred at the end of the Archean. It is unclear from the text alone whether the authors are referring to a change during the Neoproterozoic (the latter stages of the Archean) or after the Archean. This becomes problematic because the Archean data essentially show no change in the mean. A simple t-test shows that when comparing the 2.5-2.7 Ga data with the >2.7 Ga data the p-value is 0.9. In contrast the difference between the Archean data and early Proterozoic (1.8-2.5 Ga) data yields a p-value of 0.0001. This indicates that the change didn't happen during the Archean, but during the early Proterozoic. This may seem like a minor detail, but defining when these changes occurred implies geodynamic consequences that should be tied to the geologic record and geochronologic frameworks.

The statement "Initiation of plate tectonics on Earth happened in the Archean but the style of the earliest Archean tectonics may have been very different to today" does not encapsulate the raging debate on the subject. I would recommend softening the tone of the definitive "initiation of plate tectonics happened in the Archean".

Likewise, the statement "though rudimentary on-off plate tectonic events may have existed earlier from data on shales and granites" is vague. It is not clear how data from shales and granites provides insight into the onset of plate tectonics. In this case, I presume the authors are speaking of isotopic recycling generally associated with subduction processes. I would argue that plate tectonics and subduction are not the same things and isotopic recycling can happen in the absence of a global interconnected network of plates.

The issue of "modern-style" plate tectonics is far more nuanced than simply when peridotite altered by seawater was recycled back into the mantle. While I agree with the authors that the early Proterozoic (NOT the end of the Archean) ushered in the beginning of "modern-style" plate tectonics, the authors should be very clear what they mean by this. As mentioned above, simply relying on isotopic recycling is probably insufficient to take the leap to plate tectonics. A more measured way of saying this is that the effectiveness of element recycling increased substantially during the early Proterozoic that was likely

associated with more efficient subduction of altered oceanic lithosphere and MAY have been associated with a tectonic regime more akin to the modern setting.

Figure 4: I would like to see the distribution of $\delta^{18}\text{O}$ in zircon better represented rather than simply a moving mean. A moving mean fails to show the range of $\delta^{18}\text{O}$ values and how the span of data changes through time. Additionally, the reference is incorrect as Spencer et al. (2019 GCA) do not present a global database. The most up-to-date zircon $\delta^{18}\text{O}$ database is Spencer et al. (2022 EPSL). In addition to the newly compiled database, they demonstrate a decoupling in the compiled igneous records in comparison with the detrital records. Although this difference is unlikely to play a role in this current work, it will be important to specify which parts of the zircon record are being addressed in the current work.

Figure A2: This figure look as though it was copied directly out of Excel without a great deal of care (poor alignment and no axes labels).

Chris Spencer

Kingston, Ontario

8 February 2022

Reviewer #3 (Remarks to the Author):

See attachment

Review for manuscript NCOMMS-22-03547-T by Bindeman et al.:

“Oxygen isotope ($\delta^{18}\text{O}$, $\Delta^{17}\text{O}$) insights into continental mantle evolution since the Archean”

The manuscript by Bindeman et al. presents an impressive and comprehensive oxygen isotope dataset for minerals in mantle peridotite xenoliths. Based on the dataset, it is proposed that the $\delta^{18}\text{O}$ of the continental lithospheric mantle would have slightly increased since the onset of plate tectonics. If this hypothesis holds, the work would present a major advance in Earth sciences.

At present, I am not fully convinced that the interpretation of the authors, i.e., that the $\delta^{18}\text{O}$ of the continental lithospheric mantle increased through time, is fully supported by the data. My hesitance comes forth from two aspects of the study that are further outlined below. First, the statistical handling of the data by t-tests is critical to the data interpretation, but seems inadequately done in the current version of the manuscript. Second, in the current version of the manuscript, geological differences between the samples of different age groups are underexplored as a possible origin for the different $\delta^{18}\text{O}$ values of these groups. If the authors can improve these aspects of their study in a revised manuscript, then I would recommend the work to be published in *Nature Communications*, provided that the original hypothesis holds.

Please find below two major comments, followed by several minor comments on the manuscript.

Major comment 1: Data handling

In order to investigate whether mantle peridotites from different age groups have distinct $\delta^{18}\text{O}$ values or not, the authors performed t-tests on their dataset. This approach is valid, but the population sizes for the t-tests can affect their outcomes; and these seem to be chosen inadequately in the manuscript. For example, in t-tests involving mantle nodules of Proterozoic ages, the authors assumed that $n = 70$. This number includes data for replicate measurements of in total only $n = 15$ samples. These 15 samples, in turn, represent only 6 distinct Proterozoic mantle domains. As the authors intend to study the differences in $\delta^{18}\text{O}$ values between mantle domains of different ages, the t-test should consider a population of $n = 6$ in this example, and not of $n = 70$. Likewise, for Archean and Phanerozoic mantle nodules, the t-tests should be based on $n = 4$ and $n = 2$ individual sample locations, respectively. Instead, the authors calculated t-tests for population sizes of $n = 20$ and $n = 60$ analyses, respectively.

The way in which the t-tests are performed in the manuscript is important, because it may affect the conclusions in the manuscript. By using the preferred sample population sizes that are given above, in conjunction with their average $\delta^{18}\text{O}$ values, I could reproduce a significant difference between the $\delta^{18}\text{O}$ of Archean versus Proterozoic mantle peridotite, but not of Proterozoic versus Phanerozoic mantle peridotite. Altogether, I therefore recommend the authors to carefully iterate the statistical analysis of their dataset, before resubmitting their work.

I also would like to point out that the population sizes of the t-tests that are currently given in Table A4 are inconsistent with the dataset in Table A1. The number of Archean olivine samples that is stated in Table A4 ($n = 20$), for example, does not correspond to the number of samples that are listed in Table A1 ($n = 23$). Likewise, some of the average $\delta^{18}\text{O}$ values that are reported in Table A4 are different from the average $\delta^{18}\text{O}$ values in Table A1. For example: I cannot reproduce the $\delta^{18}\text{O}$ value of Archean olivine that is reported in Table A4, based on the data in Table A1. Before the work can be published, such inconsistencies between the data tables and the t-tests need to be polished out.

Major comment 2: Sample selection

The manuscript reports $\delta^{18}\text{O}$ data for mantle peridotite not only from different age groups, but also from different geological settings. For some age groups, however, samples from a given geological setting can be overrepresented. This may have introduced a bias to the interpretation of the oxygen isotope dataset, and this possibility should be discussed in the manuscript.

The Archean sample suites that were studied comprise of xenoliths that were exclusively erupted in kimberlites. Kimberlite-hosted xenoliths, however, can have strong metasomatic effects in their major and trace element concentrations. It is therefore conceivable that the elevated $\delta^{18}\text{O}$ values of the Archean peridotite samples compared to Proterozoic peridotite could reflect the composition of unusual and metasomatized cratonic mantle underneath kimberlite fields, rather than that of pristine Archean mantle. Likewise, the suite of Phanerozoic peridotite samples comprises only mantle xenoliths from arc settings. The mantle wedge above subduction zones can be intensely fluxed by fluids; and these samples are therefore not representative for typical continental lithospheric mantle in the Phanerozoic. It could therefore be argued, for example, that the distinct oxygen isotope compositions of the Archean kimberlite-hosted xenoliths compared to Phanerozoic arc-hosted xenoliths may reflect different styles of mantle metasomatism (e.g., different temperatures, fluid compositions) in the cratonic mantle below kimberlite pipes, versus metasomatism in the suprasubduction mantle wedge; rather than the onset of plate tectonics.

The authors may have arguments against strong metasomatism of their kimberlite samples. If so, I believe that a revised version of the manuscript would benefit if such arguments would be included in it.

Minor comments on the manuscript

Before commenting on the manuscript, I'd like to point out that the supplementary materials are in a very premature state for publication. Here are some examples of very simple, but absolutely important points to improve:

- Table A1 has comments written in column AA such as "To rerun (...)" that are clearly intended for the authors themselves, and not for publication.
- In Table A3, the cells have multiple font sizes, but should be uniform.
- The model ages in Table A1 column Y are currently given as dimensionless numbers, rather than in Ga (likewise in Table A3 column I).
- Some of the graphs are copy-pasted from Excel, and, for publication in a Nature journal, these graphs would benefit from a cosmetic touch-up.
-

Some detailed comments on the manuscript:

- *Title and abstract.* The title of the manuscript suggests that the conclusions in the paper will be based not only on $\delta^{18}\text{O}$ data, but also on $\Delta^{17}\text{O}$ data. The abstract, in contrast, only discusses $\delta^{18}\text{O}$ data (which I agree are the main selling point of the manuscript). For consistency, I recommend the authors either to discuss in the abstract how their $\Delta^{17}\text{O}$ data contributed to the conclusions in the manuscript or, alternatively, to delete " $\Delta^{17}\text{O}$ " from the title.

- When reporting $\Delta^{17}\text{O}$ values, the authors used apostrophe (') symbols. I believe that prime symbols (′) would be correct to use instead.
- *Introduction*: There is convincing evidence that high- $\delta^{18}\text{O}$ materials are recycled in the mantle sources of MORBs today (Eiler et al. 2000, *Nature*). This could be a good reference here.
- Lines 59-60: “*It follows, therefore, that the Archean mantle, similar to lunar rocks, should have been less affected by subducting slabs, thus more closely reflecting the BSE.*” I agree with the authors, but it needs to be discussed in the manuscript whether kimberlite-hosted Archean xenoliths, which are atypical for mantle rocks, can be used to test this hypothesis.
- Lines 81-81: “*Laser fluorination, with its error on the order of 0.01 to 0.08‰, continues to be the most precise method to resolve small 0.1-0.3‰ differences in $\delta^{18}\text{O}$.*” I recommend adding few words to this sentence, e.g., “for anhydrous silicate minerals”. Also, I think that the term “error” in this context is incorrect and that e.g., “measurement precisions” would be better.
- Lines 136 – 144: A reconstructed $\Delta^{17}\text{O}$ value of Archean mantle olivine was published by Peters et al. (2021, *Geochemical Perspectives Letters*) in a study that also includes laser fluorination $\delta^{18}\text{O}$ data for mantle peridotite. This work should be considered here.
- Lines 214 – 216: “*This computation assumes that the remaining 1-2 wt% of H_2O is released from the slab into the overlying peridotite as a low- $\delta^{18}\text{O}$ fluid (0 to +2‰) due to the breakdown of serpentine and chlorite ~100-200 km below the mantle wedge. This estimate is likely robust within a factor of 2 to 3 depending on the choice of water content, its isotopic composition, and total volume of the mantle affected*”.

This is an important thought experiment to make, and it would be nice to see the computation improved a little. I agree that fluids with $\delta^{18}\text{O}$ values as low as 0-2 ‰ could be realized by serpentine and chlorite dehydration; however, such fluids would form in the internal parts of the subducted slab, and therefore react with the upper, high- $\delta^{18}\text{O}$ parts of the slab during their ascend into the continental lithospheric mantle. It would therefore be difficult to envisage fluids with such low $\delta^{18}\text{O}$ values to react with mantle peridotite. Also, I do not fully understand how this computation would work with the information that is currently given in the manuscript. I assume that the volume of H_2O that is released into the mantle does not only depend on spreading rates, but also on the thickness of the subducted slab and on oceanic/continental crust ratios. It would be helpful to have some additional information here.

- *Fig. 1a and Fig. 1b*. The authors may want to consider plotting error envelopes on the regressions, in order to demonstrate whether the slope of the regressions are indeed positive within uncertainty or not.

Answers to Reviewers

Bigger changes.

Per the Editor suggestion and since we have more space in the Nature Communications paper, we have brought two Figures from the Supplementary (Fig. A1 and 2) into the main text with associated text and discussion. We have also increased the length of the Introduction to put our dataset in a bigger context as well as we increased Discussion and number of references. The paper now has 6 Figures and 4500 words in the main text, in line with other Nature Communications papers.

Reviewer #1 (Remarks to the Author):

Review NCOMMS-22-03547-T

Author: Ilya Bindeman

Fig. 1. With the exception of 1 data point at 2.1 Ga, a single straight line can be fit through the data, with no bump at 2.5, so it is hard to interpret this data as a sudden start of tectonics at 2.5 Ga, with none before.

>> We agree, the language was softened a bit, although the bar diagram does show a bit of the step at 2.5-2.0 Ga and the dataset can be fit with steeper change from 3 to ~2 and then shallow tend from 2 to 0 Ga.

Abstract

>>shortened to be 150 words

Written well, but please note the end of the Archean is not 2.5-2.0 Ga, it is 2.5 Ga.

>We removed this age range and changed the sentence a bit.

The origin of the CLM is assumed to be accreted melting residues, but there are some other models and choices, and they are not all mutually exclusive. It is more complex.

>>we agree that mantle melting is complex but other mechanisms also result in melt depletion relative to primitive mantle (BSE) one way or another. In any case, our collection has residues of melt extraction in different settings (cratons, intraplate off-craton, subduction zones, continental margins) as well as an extreme case of no-melting (rocks with BSE major element composition attached to the CLM on cooling). We added 2 sentences and references to Intro.

“Formation and subsequent growth of SCM and continental keels involve imbrication upon collisions of depleted and buoyant subducted oceanic lithosphere (including melt depletion directly at subduction settings followed by slab delamination) and thickening by compression around existing continental margins or subarc settings [28, 31-32]. It can thus be expected that SCM, like Earth’s surface rocks, have different ages and details of chemical and isotopic composition related to its age and magmatic, geologic prehistory.”

So already there is a model assumption that the oldest CLM was not influenced by subduction. It should be discussed that in some views, subduction was operating since the Hadean. Maybe it was just less-influenced by shorter exposure to subduction?

>changed to: ...due to the development and progression of plate tectonics.

Our oldest samples are ~3.5 Ga, only 1 Ga after the Giant Impact, and potential plate tectonics (if at all) would have operated only 25% of the Earth’s history, if plate tectonic started right away after 4.4 Ga. We recall that the earliest Plate tectonics has been suggested is 3.9 Ga (Bedard, 2018 for review).

So the formulation above is Ok we think.

BSE abbreviation is used before it is defined

>>moved definition up

Line 25. Yes, from looking at the data plots, it seems to be a general progression (nearly linear) with no real obvious jump at the Archean / Proterozoic boundary. From examining the data and figures first, before the text, I feel perhaps that the authors are using one of the popular models of the initiation of plate tectonics (at 2.5 Ga) and trying to make their data fit that model, but only considering the data from O isotopes. But, lets see. Topic is interesting.

>>We soften the language a bit with respect to the “window of change”. We also added a statement that the timing of initiation of plate tectonics is a contentious topic. No, we do not need to make any assumptions here about the timing of plate tectonics, nor do we have a preferred model! Another option could be that that the plate tectonic regime, initiated at ~2.5 Ga, coexisted with other tectonic regimes for a while, which could produce a smoother, gradual O-isotope transition. We recognize subtle d18O changes and do not interpret second order trends.

Introduction

Line 50-55. Assuming, taking for granted that plate tectonics started in the Archean, or as here, at the Archean Proterozoic boundary, is a highly debated issue, and one of the most contentious in Earth Sciences. Only one reference is given, and it is one that is widely debated by the geological community, but perhaps it fits the model the authors are trying to support. At the very least the authors must discuss this issue, and bring up the various possibilities, and then test if their data supports any of the models. As it stands now, this is a fatal flaw in the manuscript, to assume this is correct, then take a gradual trend in the O data, and say it indicates plate tectonics started at 2.5 Ga.

>>We rephrased these sentences here to discuss more possibilities and added references, including the ones below. No, we do not need to make any assumptions here about timing! We also added text and references to discussion

If that is so, the data must be consistent with all of the other evidence, geological, geochemical, and so on, much of which indicates a much earlier start to some form of subduction and plate tectonics. Please see recent reviews by Harrison 2020 (<https://dx.doi.org/10.1007/978-3-030-46687-9>), Keller and Harrison in PNAS (<https://doi.org/10.1073/pnas.2009431117>), and Windley in PR (doi.org/10.1016/j.precamres.2020.105980). If the starting point is not assuming that plate tectonics started at 2.5 Ga, then the authors could make a more scientific analysis and might find something different, perhaps (as the data suggests to me) a slow gradual change in the O isotopes induced by a steady stream of subduction since the earliest records.

>>We agree, we are not making any assumption nor do we have anything at stake to suggest one way or another. We observe gradual change since 3.0Ga. We also added suggested references.

As for “sagduction”, that phrase is popular in some circles, but the process is poorly defined and has never been observed or proven anywhere.

>>added “vertical lithospheric delamination”

Why are the xenoliths of “tentative” Archean age?

>> removed “tentative”, they are Archean in age as determined by Re-Os TRD

65-70. As mentioned above, there are also several models for the formation of lithospheric keels, not just

this simple one you list here. To be fair, you should discuss them, and discuss if your O isotopes could tell the difference. What about the evidence from Shirey and Richardson for subduction back to at least 3.1 Ga based on the isotopes of eclogitic and peridotitic inclusions? What about the models of keel formation by slab stacking?

Added:

...and perhaps as early as 3.0-3.2 Ga (Shirey and Richardson, 2011) when eclogitic diamonds became prevalent over peridotitic.

We cite a common scenario.

- Slab stacking at 3 Ga is not consistent with the composition of modern slab CLM, e.g. in western Pacific, like our Avacha xenoliths, which has much lower Mg# (<0.92, due to low-pressure melt extraction) than typical cartoon mantle (>0.92).

I think this all needs to be reframed to test these models, not to assume one of the most controversial models is correct, then say your data supports it, when there is barely any change at the moment you say the planet suddenly changed from a non-plate tectonic world to a plate tectonic world with subduction recycling.

>>Good overall suggestion that helped us streamline Intro, we added references to these points. We do not assume (or need to) that plate tectonic started at exactly 2.5 Ga. We also cite new references on slightly older plate tectonic initiation (e.g. Shirey and Richardson, 2011) and models of continental root formation (see above). While reviewing the literature on the topic, the broad consensus for relatively modern style plate tectonic with its features such as paired metamorphic belts, the chemistry of magmas, supercontinent cycle, and others) is in the range of 3.2 (sometimes 3.5) Ga to 2.1 Ga. Very few publications push it to Neoproterozoic (Stern et al 2018) or Hadean. For the purpose of our study, a change occurring at 2.0-3.0 (3.2) Ga is sufficient to explain the data.

We inserted for example:

“Other two common scenarios of formation of CLM roots involve depletion during MORB production, then successive imbrication upon collisions of buoyant subducted oceanic lithosphere, or melt depletion directly at subduction settings followed by slab delamination and thickening by compression in around existing continental margins or subarc settings (Canil and Lee, 2009; Herzberg and Rudnick, 2012; Wang et al. 2016). “

Results

The data and the discussion of the data are all fine. So the data show a gradual decrease in O18 with time, at a fairly constant rate through the whole 3 billion years of the data sampling period.

For the Archean, you may do better to discuss the Neoarchean, Mesoarchean, and Paleoarchean, since the mantle temperature and melting degree surely changed a lot through those 1.5 billion years. By stating the change this way, as the change from Archean/Proterozoic, the authors are giving the readers the wrong impression that something suddenly happened. This is not consistent with the data shown, or with many other data sets not considered in this analysis.

>>Our oldest samples are 2.9-3.0 Ga, so Neoarchean. We added the word “gradual” to the “temporal change”. In the text writing, we do not now have reference to a sudden change occurring at 2.5 Ga.

Discussion

Line 170. There are peridotites of the age range of 3.0 – 3.5, and older. I am not sure why the authors say there is a lack of olivine formed at that time. Maybe say you have no data on samples from that age.

We don't have solid-evidence 3.5 Ga peridotites in our collection. The oldest Re-dated xenoliths are in the 3.0-3.5 Ga time window in the Kaapvaal craton. The ages >3 Ga are based on measurements of sulfides in peridotites, not bulk rocks, hence can be interpreted otherwise than rock formation ages. Perhaps more importantly, many xenoliths in S Africa are altered (weathered) and thus $\delta^{18}\text{O}$ analyses are often impossible. But yes we have 3.0Ga samples.

I am uncomfortable with the assumption here, of using the O from olivine from younger than 3.0-3.5 Ga, and using the O isotopes from zircon for 3.0-3.5 and older, to derive a single curve (line).

>> We did not use zircons to derive the line, line is only based on olivines and bulk. Zircons are just plotted and show that they are on the extension of the line. We now have a discussion of isotopic fractionations between zircons and olivines, peridotites, and basalts.

Summary

The data presented is of high quality, and shows a gradual trend over the course of 3 Ga. This could be related to continuous subduction over 3 Ga.

>> yes, we agree

The authors have made too many assumptions and simply accept that plate tectonics started at the Archean/Proterozoic boundary, instead of using their data to test that hypothesis. If they reframed their argument as a test of this hypothesis and linked better with other data sets, both geochemical and geological, I think they would have a stronger paper, and a different answer.

>>see above how we modified the intro and also Discussion about the initiation of plate tectonics. It was never our goal to say that it started at 2.5 Ga! We are just comparing Archean vs Proterozoic vs Phanerozoic groups of samples.
Hope the new draft satisfies the Reviewer's comments.

Reviewer #2 (Remarks to the Author):

Dear Editor,

Below is my review of Oxygen isotope ($\delta^{18}\text{O}$, $\Delta^{17}\text{O}$) insights into continental mantle evolution since the Archean by Bindeman et al., submitted to Nature Communications. This manuscript provides convincing evidence for a decrease in the $\delta^{18}\text{O}$ composition of the continental lithospheric mantle as probed by crustal xenoliths. Below are some minor suggestions on how this manuscript can be improved and some of their main points clarified. Although the paper is well written, there are a number of places where the verbiage is vague and imprecise. Nevertheless and notwithstanding my longwinded comments below, I see no reason why this paper cannot be published with minor revisions.

Title/Abstract: The title highlights the $\delta^{18}\text{O}$ and $\Delta^{17}\text{O}$, but the abstract doesn't discuss the $\Delta^{17}\text{O}$ at all. The $\Delta^{17}\text{O}$ shows minor variation in comparison to the $\delta^{18}\text{O}$. This is a significant finding in the light of previous $\Delta^{17}\text{O}$ work showing shifts in $\Delta^{17}\text{O}$ of shale and granite through time. The absence of any

change in $\Delta^{17}\text{O}$, while the $\delta^{18}\text{O}$ does show a shift, should be highlighted in the abstract.

>> Done added a short sentence in Abstract about $\Delta^{17}\text{O}$

The paper focuses a lot on changes that occurred at the end of the Archean. It is unclear from the text alone whether the authors are referring to a change during the Neoproterozoic (the latter stages of the Archean) or after the Archean. This becomes problematic because the Archean data essentially show no change in the mean. A simple t-test shows that when comparing the 2.5-2.7 Ga data with the >2.7 Ga data the p-value is 0.9. In contrast the difference between the Archean data and early Proterozoic (1.8-2.5 Ga) data yields a p-value of 0.0001. This indicates that the change didn't happen during the Archean, but during the early Proterozoic. This may seem like a minor detail, but defining when these changes occurred implies geodynamic consequences that should be tied to the geologic record and geochronologic frameworks.

>> We agree, we now just say that there is a decreasing trend through time, this is the safest conclusion one can make. We refrain from interpreting second-order structure to this trend or assigning exact ages to transition. see also the same answer to Rev 1 above

The statement "Initiation of plate tectonics on Earth happened in the Archean but the style of the earliest Archean tectonics may have been very different to today" does not encapsulate the raging debate on the subject. I would recommend softening the tone of the definitive "initiation of plate tectonics happened in the Archean".

>> We modified these two sentences and also added new references and text including models of continental mantle formation.

Likewise, the statement "though rudimentary on-off plate tectonic events may have existed earlier from data on shales and granites" is vague. It is not clear how data from shales and granites provides insight into the onset of plate tectonics. In this case, I presume the authors are speaking of isotopic recycling generally associated with subduction processes. I would argue that plate tectonics and subduction are not the same things and isotopic recycling can happen in the absence of a global interconnected network of plates.

>> We completely removed this sentence, and say now that the observed trend "...support a gradual cumulative and putative effects of supplying surface derived material into the CLM"

The issue of "modern-style" plate tectonics is far more nuanced than simply when peridotite altered by seawater was recycled back into the mantle.

>> see above we removed the sentence and added more text and references in the Introduction.

While I agree with the authors that the early Proterozoic (NOT the end of the Archean) ushered in the beginning of "modern-style" plate tectonics, the authors should be very clear what they mean by this. As mentioned above, simply relying on isotopic recycling is probably insufficient to take the leap to plate tectonics. A more measured way of saying this is that the effectiveness of element recycling increased substantially during the early Proterozoic that was likely associated with more efficient subduction of altered oceanic lithosphere and MAY have been associated with a tectonic regime more akin to the modern setting.

>> Good suggestion we rephased this the way suggested. We added more text and references in the Introduction, but Rev. 1 wanted us to accept the beginning of plate tectonic was 3.5 Ga. We now describe a gradual trend without making statements about the start of plate tectonics.

Figure 4: I would like to see the distribution of $\delta^{18}\text{O}$ in zircon better represented rather than simply a

moving mean. A moving mean fails to show the range of $\delta^{18}\text{O}$ values and how the span of data changes through time. Additionally, the reference is incorrect as Spencer et al. (2019 GCA) do not present a global database. The most up-to-date zircon $\delta^{18}\text{O}$ database is Spencer et al. (2022 EPSL).

>>We replaced the reference of concurrently appeared paper by Spencer et al 2022. Our paper was first submitted on March 9, 2021, so we were unaware. With respect to showing ranges vs a line, this would make the figure too cumbersome, we would prefer to have a moving mean of all zircons (Spencer 2020 database).

In addition to the newly compiled database, they demonstrate a decoupling in the compiled igneous records in comparison with the detrital records. Although this difference is unlikely to play a role in this current work, it will be important to specify which parts of the zircon record are being addressed in the current work.

>>We show all zircon data (detrital and igneous) for simplicity as they both relate to “crust” which we try to portray. Our paper is about the mantle and other reservoirs (shales, granites, zircons) are shown for reference.

Figure A2: This figure look as though it was copied directly out of Excel without a great deal of care (poor alignment and no axes labels).

>>All Figures were improved and two were moved to the main text.

Chris Spencer
Kingston, Ontario
8 February 2022

>>Thank you for your careful review

Rev 3

The manuscript by Bindeman et al. presents an impressive and comprehensive oxygen isotope dataset for minerals in mantle peridotite xenoliths. Based on the dataset, it is proposed that the $\delta^{18}\text{O}$ of the continental lithospheric mantle would have slightly increased since the onset of plate tectonics. If this hypothesis holds, the work would present a major advance in Earth sciences. At present, I am not fully convinced that the interpretation of the authors, i.e., that the $\delta^{18}\text{O}$ of the continental lithospheric mantle increased through time, is fully supported by the data. My hesitance comes forth from two aspects of the study that are further outlined below. First, the statistical handling of the data by t-tests is critical to the data interpretation, but seems inadequately done in the current version of the manuscript.

>> We now have performed line fit error analysis, sample grouping by age and t-test, and splitting the dataset by locality (new Fig. 2). All demonstrate that trends hold and some r^2 improved! For example, splitting the data by locality results in r^2 of 0.46 to 0.52. We attach spreadsheets with these calculations. These results are presented in Tables A4 and A5

Second, in the current version of the manuscript, geological differences between the samples of different age groups are underexplored as a possible origin for the different $\delta^{18}\text{O}$ values of these groups. If the authors can improve these aspects of their study in a revised manuscript, then I would recommend the work to be published in Nature Communications, provided that the original hypothesis holds.

>> Coauthor Dmitry Ionov (from whom the majority of the collection was obtained over the years, and a lead specialist in mantle xenolith geochemistry) replies: 1) Xenoliths in kimberlites are the ONLY whole-

rock mantle samples that yield AR ages (the meaning of Re-Os dates on individual mineral grains is unclear). (2) Nearly all our samples experienced no or little metasomatism. In general, the metasomatism in common (“coarse”) cratonic CLM peridotites in this study does not affect major elements, but concerns only minor and trace elements that are negligible in WR mass balance. For instance, major elements, e.g. Mg isotopes of modern arc mantle (Hu et al., 2021, GCA), as well as of cratonic CLM, are not affected by metasomatism. The only cratonic CLM samples that show major element metasomatism are those from the CLM base, >180 km; these are not suitable for Re-Os dating and are NOT present in this study.

We added the following text in the ms (end of Introduction):

“These samples are not uniformly distributed geographically because explosive volcanic eruptions: kimberlites and alkali basalts that fragment the CLM and can carry mantle xenoliths to the surface, are restricted to volatile-rich magmas. These, however, span many tectonic settings and regions on Earth, and occur above hot spots, at continental plate boundaries, and rift zones where explosive eruptions rapidly carry pieces of the mantle to the surface at rates of meters per second (e.g. [42]). This rapid transport prevents extensive chemical and isotopic exchange between the host melt and the xenoliths for the majority, and especially major elements including oxygen. Finally, CLM fragments are much easier to find in, and extract from, loose pyroclastic facies of volcanic eruptions than in massive (and slower cooled) lavas. This is why, much work on mantle xenoliths is done on the best available samples at each locality worldwide, and in specific regions. In such cases, the work is typically performed on the largest, freshest, or least altered mantle xenoliths from several such sites. Comparison with smaller, partially altered, but more abundant xenoliths from broader regions usually indicate the selected best samples are representative of larger areas in a specific lithospheric block and tectonic setting ([34, 36-42 and references therein). This statement characterizes the suite of xenoliths we present here. “

Please find below two major comments, followed by several minor comments on the manuscript.

Major comment 1: Data handling

In order to investigate whether mantle peridotites from different age groups have distinct $d_{18}O$ values or not, the authors performed t-tests on their dataset. This approach is valid, but the population sizes for the t-tests can affect their outcomes; and these seem to be chosen inadequately in the manuscript. For example, in t-tests involving mantle nodules of Proterozoic ages, the authors assumed that $n = 70$. This number includes data for replicate measurements of in total only $n = 15$ samples. These 15 samples, in turn, represent only 6 distinct Proterozoic mantle domains. As the authors intend to study the differences in $d_{18}O$ values between mantle domains of different ages, the t-test should consider a population of $n = 6$ in this example, and not of $n = 70$. Likewise, for Archean and Phanerozoic mantle nodules, the t-tests should be based on $n = 4$ and $n = 2$ individual sample locations, respectively. Instead, the authors calculated t-tests for population sizes of $n = 20$ and $n = 60$ analyses, respectively. The way in which the t-tests are performed in the manuscript is important, because it may affect the conclusions in the manuscript. By using the preferred sample population sizes that are given above, in conjunction with their average $d_{18}O$ values, I could reproduce a significant difference between the $d_{18}O$ of Archean versus Proterozoic mantle peridotite, but not of Proterozoic versus Phanerozoic mantle peridotite. Altogether, I therefore recommend the authors to carefully iterate the statistical analysis of their dataset, before resubmitting their work.

>>We now added lines to a Table in the Supplementary splitting of the dataset as recommended, by the average value of each nodule (n=104), by the average value of each locality (keeping AR and PR dated and compositionally distinct Udachnaya and Obnazhennaya samples as separate groups, total n=15, new Fig. 2), and splitting them into 3-2.0 vs 1.9-0 Ga groups.

We also performed the line fit statistics error analysis and present these in Table A5, mentioning it in the text and in new Fig 2. Splitting data as above actually improves our trends.

I also would like to point out that the population sizes of the t-tests that are currently given in Table A4 are inconsistent with the dataset in Table A1. The number of Archean olivine samples that are stated in Table A4 (n = 20), for example, does not correspond to the number of samples that are listed in Table A1 (n = 23). Likewise, some of the average d18O values that are reported in Table A4 are different from the average d18O values in Table A1. For example: I cannot reproduce the d18O value of Archean olivine that is reported in Table A4, based on the data in Table A1. Before the work can be published, such inconsistencies between the data tables and the t-tests need to be polished out.

>>Thank you, these are results added more data last minute. We checked consistency now.

Major comment 2: Sample selection

The manuscript reports d18O data for mantle peridotite not only from different age groups but also from different geological settings. For some age groups, however, samples from a given geological setting can be overrepresented. This may have introduced a bias to the interpretation of the oxygen isotope dataset, and this possibility should be discussed in the manuscript.

>>We added a sentence to the Intro that we report all analyses of peridotites that came through the lab over 13 years for different purposes, thus that was random sampling. Also, we saw a similar trend in Matthey et al dataset. We plotted the dataset by locality (Fig. 2), thus removing overrepresented samples. This action actually improved the trends.

The Archean sample suites that were studied comprise of xenoliths that were exclusively erupted in kimberlites. Kimberlite-hosted xenoliths, however, can have strong metasomatic effects in their major and trace element concentrations. It is therefore conceivable that the elevated d18O values of the Archean peridotite samples compared to Proterozoic peridotite could reflect the composition of unusual and metasomatized cratonic mantle underneath kimberlite fields, rather than that of pristine Archean mantle.

>> Coauthor D. Ionov (from whom the majority of the collection was obtained over the years, and a lead specialist in mantle xenolith geochemistry) replies: 1) Xenoliths in kimberlites are the ONLY whole-rock mantle samples that yield AR ages (the meaning of Re-Os dates on individual mineral grains is unclear). (2) Nearly all our samples experienced no or little metasomatism. If present, the metasomatism in common (“coarse”) cratonic CLM peridotites in this study does not affect major elements, but concerns only minor and trace elements that are negligible in WR mass balance. For instance, major elements, e.g. Mg isotopes of modern arc mantle (Hu et al., 2021, GCA), as well as of cratonic CLM, are not affected by metasomatism. The only cratonic CLM samples that show major element metasomatism are those from the CLM base, >180 km; these are not suitable for Re-Os dating and are NOT present in this study. (3) Our collection of kimberlite-hosted xenoliths yields both AR and PR ages. These ages are related to rock types and tectonic setting in individual cratons, not degrees of metasomatism. (4) Our xenoliths from the Siberian craton are among the least metasomatized (and least altered) worldwide, for instance, they almost never contain volatile-bearing mica or amphibole, unlike xenoliths from the Kaapvaal craton.

See also our new paragraph at the end of Intro quoted above.

Likewise, the suite of Phanerozoic peridotite samples compasses only mantle xenoliths from arc settings.

>>No, actually the majority are from intra-continental regions with Paleozoic to late Proterozoic formation ages (Mongolia, southern Siberia)

The mantle wedge above subduction zones can be intensely fluxed by fluids, and these samples are therefore not representative for the typical continental lithospheric mantle in the Phanerozoic.

>> See the previous comment that we have a diverse set of Phanerozoic xenoliths. Major and trace elements, as well as heavy stable isotope (Mg, Fe...), data on our Kamchatka xenoliths, show only minor effects on trace elements, and no significant effects on major elements in subduction settings.

It could therefore be argued, for example, that the distinct oxygen isotope compositions of the Archean kimberlite-hosted xenoliths compared to Phanerozoic arc-hosted xenoliths may reflect different styles of mantle metasomatism (e.g., different temperatures, fluid compositions) in the cratonic mantle below kimberlite pipes, versus metasomatism in the suprasubduction mantle wedge; rather than the onset of plate tectonics.

>> A major point in this paper is that oxygen, and other major oxides, are unaffected by metasomatism (see our data and those of Matthey et al), because of mass balance consideration (too much fluid required relative to the CLM mass) to change oxygen. Also, our Phanerozoic xenolith set is not just W Bismarck and Kamchatka samples, and Archean samples are only available from kimberlites worldwide.

The authors may have arguments against the strong metasomatism of their kimberlite samples. If so, I believe that a revised version of the manuscript would benefit if such arguments would be included in it.

Coauthor D. Ionov replies: - The arguments for the absence of effects of metasomatism on major elements (Mg, Fe, Al...), for instance, that their relations are fully consistent with an origin as melting residues, are provided in original papers that reported and described our samples (Carlson and Ionov, 2019; Ionov et al. 2010, a few others, all of which are listed in the Supplementary).

But also see the added Intro paragraph above how xenoliths are selected and studied.

Minor comments on the manuscript

Before commenting on the manuscript, I'd like to point out that the supplementary materials are in a very premature state for publication. Here are some examples of very simple, but absolutely important points to improve:

- Table A1 has comments written in column AA such as "To rerun (...)" that are clearly intended for the authors themselves, and not for publication.

Table A1 was formatted better and a number of significant figures in each column were made uniform, the same font was used. A total number of analyses, the sample analyzed, and averages, were recounted.

>>Supplementary Tables have been checked and checked again removing these comments

- In Table A3, the cells have multiple font sizes but should be uniform.

>>done

- The model ages in Table A1 column Y are currently given as dimensionless

>>done

-numbers, rather than in Ga (likewise in Table A3 column I).

>>corrected

Some of the graphs are copy-pasted from Excel, and, for publication in a Nature journal, these graphs would benefit from a cosmetic touch-up.

>>improved

Some detailed comments on the manuscript:

- **Title and abstract.** The title of the manuscript suggests that the conclusions in the paper will be based not only on d18O data but also on D'17O data. The abstract, in contrast, only discusses d18O data (which I agree

are the main selling point of the manuscript). For consistency, I recommend the authors either to discuss in the abstract how their D'17O data contributed to the conclusions in the manuscript or, alternatively, to delete "D'17O" from the title.

>>This is also recommended by Rev1. We inserted in the Abstract that D17O is constant.

- When reporting D'17O values, the authors used apostrophe (') symbols. I believe that prime symbols (′) would be correct to use instead.

>>done

- **Introduction:** There is convincing evidence that high-d18O materials are recycled in the mantle sources of MORBs today (Eiler et al. 2000, Nature). This could be a good reference here.

>>added to the reference list

- Lines 59-60: "It follows, therefore, that the Archean mantle, similar to lunar rocks, should have been less affected by subducting slabs, thus more closely reflecting the BSE." I agree with the authors, but it needs to be discussed in the manuscript whether kimberlite-hosted Archean xenoliths, which are atypical for mantle rocks, can be used to test this hypothesis.

(1) No other AR rocks are available.

(2) See above, they are consistent with an origin by melt extraction, with metasomatism, if present, only affecting incompatible trace elements.

- Lines 81-81: "Laser fluorination, with its error on the order of 0.01 to 0.08‰, continues to be the most precise method to resolve small 0.1-0.3‰ differences in d18O. " I recommend adding a few words to this sentence, e.g., "for anhydrous silicate minerals". Also, I think that the term "error" in this context is incorrect and that e.g., "measurement precisions" would be better.

>>why only anhydrous? Some hydrous glasses and amphiboles are reproduced even better than olivines.

- Lines 136 – 144 A reconstructed D'17O value of Archean mantle olivine was published by Peters et al. (2021, Geochemical Perspectives Letters) in a study that also includes laser fluorination d18O data for mantle peridotite. This work should be considered here.

>>added this reference and a short statement that our data agrees with his on D17O. They do not have many d18O analyses as we do.

- Lines 214 – 216: "This computation assumes that the remaining 1-2 wt% of H2O is released from the slab into the overlying peridotite as a low-δ18O fluid (0 to +2‰) due to the breakdown of serpentine and chlorite ~100-200 km below the mantle wedge. This estimate is likely robust within a factor of 2 to 3 depending on the choice of water content, its isotopic composition, and total volume of the mantle affected".

This is an important thought experiment to make, and it would be nice to see the computation improved a little. I agree that fluids with δ18O values as low as 0-2 ‰ could be realized by serpentine and chlorite dehydration; however, such fluids would form in the internal parts of the subducted slab, and therefore react with the upper, high-δ18O parts of the slab during their ascent into the continental lithospheric mantle. It would therefore be difficult to envisage fluids with such low δ18O values to react with mantle

peridotite. Also, I do not fully understand how this computation would work with the information that is currently given in the manuscript. I assume that the volume of H₂O that is released into the mantle does not only depend on spreading rates, but also on the thickness of the subducted slab and on oceanic/continental crust ratios. It would be helpful to have some additional information here.

>>we assumed 100 km CLM peridotite thickness affected by the upper 10 km of devolatilizing serpentinitic slab with low-d¹⁸O fluids, losing 1-2 wt% water with isotopic values of 0 to +2‰. Subduction zone area are 62,000 km plate subduction 8 cm/yr. Over 2 Ga, these parameters result in the release of 2.58e21 kg of water. If the water is 0‰ and is fluxed homogeneously over the entire 100 km thick d¹⁸O=5.4‰ mantle (1.65e23 kg), it will decrease the mantle to 5.32‰. Should only half of the mantle be affected, it will get 5.23, 33% will result in 5.14‰, and so forth.

But correct, two parameters were missing from the above: We added "...released from the upper 10 km of the slab into 100 km of the overlying CLM, at 62K subduction length and 8 cm/yr convergence rate, see the new text.

- Fig. 1a and Fig. 1b. The authors may want to consider plotting error envelopes on the regressions, in order to demonstrate whether the slope of the regressions are indeed positive within uncertainty or not.

>> added 95% error envelopes to Fig 1 using Datagraph program
We also performed line fit error analysis in Supplementary (Table A4) and refer to this in the text.

REVIEWER COMMENTS

Reviewer #1 (Remarks to the Author):

The work is original, and the data robust. The variations in the data are small, but convincingly significant.

The work should be of interest to readers concerned about the oxygen isotopic composition of the mantle, and upper Earth systems through time, and how that may relate to weathering and the development of plate tectonics.

The data show a trend with time, but not giant jumps or step functions.

Since this is a revision, I focused more on if the authors addressed the comments from the first reviewers, than adding new comments. In that light, my comments on the revision are below:

Response to reviewers

Reviewer 1

First few ok.

It is hard to understand the answer to the question on whether or not the oldest CLM is influenced by subduction, because of the grammar. It is strange to suggest Bedard 2018 as a reference for earliest start to subduction at 3.9 Ga, as those papers are unconstrained speculations with misrepresentations of the geological data. This was discussed in a big review paper, including the Bedard paper, by B Windley, which seems to be cited and discussed now.

For the question on slab stacking, there is a new paper in Geology 2022 by Z Wang that may be relevant.

Still, the models of Windley, Korenaga, Harrison, Wilde, others for very early start of plate tectonics seem not considered, but instead take the "consensus" view (which is not always the correct one in science).

In response to reviewers, the authors say 2.9-3.0 is Neorchean, but it is not. It is Mesoproterozoic.

Reviewer 2

The answers to the statistics on when the changes happened, as a simple trend in time, is a good solution

Reviewer 3

The authors say the statistical treatment of the data is improved

The point of further describing the geological differences between the samples with different d_{18O} values is very important. The revised text addresses these concerns as well as possible, for xenolith samples.

Manuscript.

The references are in some place listed as numbers, some by author and year, some both. Please prepare a cleaner manuscript!

Line 8-85. There is a new paper in PNAS on field examples of Archean eclogitic oceanic crust that may be relevant

Line 85-90. Please present the counter-arguments that hotter mantle does not prevent modern style plate tectonics from a physical point of view, see Jun Korenaga papers, or O Weller papers, for their models.

It seems ref 25 should be moved away from the group at line 93 calling for sagduction (they argued against that) to the references at line 90, and reference 22 be moved to line 26, for the group that argues for sagduction or other things that don't happen on the present Earth.

I see the questions from reviewers are indeed mostly addressed

Results

The data seem indeed to show a gradual trend over time, with a small deflection focused on the large number of samples (from Udachaya) at 2.0 Ga, and for the post Mesozoic set.

Line 233. Since the Archean samples sampled, are 15% more melt depleted than the post-Archean set., an additional correction is applied to the Archean data. I understand, but still find this troubling, to use the Archean boundary to apply a correction before and after it, and then to look for changes before and after the boundary.

Discussion

This is a fair discussion, and has incorporated the comments of the reviewers, so is more balanced than the original version.

Overall the manuscript is improved, and questions answered, to the degree possible with such small changes in the limited data set.

Reviewer #2 (Remarks to the Author):

Dear Editor,

I am happy with the revisions that were made and would be happy to see this published in its current form.

Best,

Chris Spencer

Kingston, Ontario

19 April 2022

Reviewer #3 (Remarks to the Author):

Resubmitted manuscript by Bindeman et al.

The manuscript by Bindeman et al. has now been improved compared to the previous version. My most important concerns with the manuscript have now been dealt with, i.e., the statistical handling of the data was been improved, the supplementary materials were polished, and sufficient context of the sample suite was given in the response letter by the authors. I recommend the work to be published, but noticed a few sentences in the abstract and introduction that need some editing, because they are not fully correct as they are given in the current version of the manuscript:

Line 12 – “Oxygen isotopic ratios are homogenous in the bulk of Earth’s mantle”. This statement is hard to prove, and recent studies on phenocrysts from ocean island basalts (including high $^3\text{He}/^4\text{He}$ basalts from Iceland) would disagree (work by Maja Rasmussen et al.). Oxygen isotope ratios are pretty homogeneous in mantle xenoliths as is pointed out later in the manuscript; but mantle xenoliths do not necessarily give a representative image of the bulk of Earth’s mantle. I’d recommend the authors to tone down this statement.

Lines 24 - 26 “ $\delta^{18}\text{O}$ of the continental lithospheric mantle has decreased by 0.2‰ since the Archean due to the initiation of plate tectonics and redistribution of low- $\delta^{18}\text{O}$ and high $\delta^{18}\text{O}$ materials in the subduction zones, explaining crust-upper mantle mass balance”. Why not write that the 0.2‰ change is possibly consistent with crust-upper mantle mass balance? I believe that the causal relationship here is arguable.

Lines 29-39: “Earth’s peridotitic mantle is the predominant reservoir of the planet, largely controlling elemental and isotopic mass balances planet-wide”. The statement is incomplete/incorrect. The mantle is the predominant reservoir for lithophile elements, but not for siderophile elements like Fe, Ni, and the HSE; nor for atmophile elements like N and the noble gases. Perhaps write “(..) predominant oxygen reservoir “

Line 58: “Oxygen is the main element in the solar system” This statement is not correct, because the Si-normalized abundances in the Solar System are much higher for H and He than for O. Perhaps exchange “solar system” with “the terrestrial planets” and the sentence would work.

May 16, 2022

Answers to Reviewers

Reviewer #1 (Remarks to the Author):

The work is original, and the data robust. The variations in the data are small, but convincingly significant.

The work should be of interest to readers concerned about the oxygen isotopic composition of the mantle, and upper Earth systems through time, and how that may relate to weathering and the development of plate tectonics.

The data show a trend with time, but not giant jumps or step functions.

Since this is a revision, I focused more on if the authors addressed the comments from the first reviewers, than adding new comments. In that light, my comments on the revision are below:

Response to reviewers

Reviewer 1

First few ok.

It is hard to understand the answer to the question on whether or not the oldest CLM is influenced by subduction, because of the grammar. It is strange to suggest Bedard 2018 as a reference for earliest start to subduction at 3.9 Ga, as those papers are unconstrained speculations with misrepresentations of the geological data. This was discussed in a big review paper, including the Bedard paper, by B Windley, which seems to be cited and discussed now.

>>we moved the citation of Bedard 2018 down the clause after “subduction” this is what he is proposing along with several other scientists (Herve Martin and JF Moyer for example). We also inserted a reference to Nick Arndt’s excellent review paper from 2013, where he discusses all aspects of the pro and cons of early vs late initiation of plate tectonics and also discusses Bedard’s model; Arndt is a proponent of “early” plate tectonics but “early” for him is 3.8 Ga. Most agree that subduction was in operation by 2.7 Ga. Our oldest xenoliths are from 3.0 Ga. Even if they have seen 0.8 Ga of subduction, this early subduction as Nick Arndt writes, was hot and dehydration of slabs happened earlier, so no low-d18O fluids that we describe in our work have significantly affected the CLM for these first 0.8 Ga as it has cumulatively affected the CLM for the next 3 Ga.

We do not know if Bedard was misrepresenting data, he is working in the Geological Society of Canada and clearly has rich field experience with respect to Precambrian geology.

It is really hard for us authors to take a side in the argument “when did plate tectonic start”. I will distract readers from the main findings of our paper.

But OK, We added the following paragraph in Results:

“Our estimate of the $\delta^{18}\text{O}_{\text{BSE}}$ is of course related to the timing of initiation of plate tectonics, which is a matter of significant debate [7,11, 22-26]. If significant subduction started much earlier than 3 Ga, and TRD ages were also reset, then the pre-plate tectonic $\delta^{18}\text{O}_{\text{CLM}}$ would have been slightly higher than we estimate here. For example, extrapolation of the linear trend for CLM in Fig. 1 would result in 5.61‰ at 3.8 Ga and 5.63‰ at 4.1 Ga, but within error of our measured Archean estimate of 5.57 ± 0.07 ‰. We notice, however, that the temporal change in $\delta^{18}\text{O}$ in the Archean samples from 3.0 to 2.6 Ga is less steep than the change between the Archean and Proterozoic and Archean vs all post-Archean samples (Fig. 1c), and so extrapolation $\delta^{18}\text{O}$ value is less than the above estimates. Slab dehydration on early hotter Earth would likely happen at shallower conditions (24) and the oxygen isotope effects that we describe below, were likely less prominent. Perhaps more importantly, the oldest peridotite samples in our collection are ~ 3 Ga and there are no CLM peridotite xenoliths with robust bulk-rock Re-Os ages significantly more than 3 Ga [30, 41]. This perhaps reflects that there may not be much left of surviving earliest Eoarchean and Hadean CLM as it was recycled back into the convecting mantle immediately after the Hadean (9,11, 24), annihilating the effects of early subduction (if any) on crust-CLM oxygen isotope repartition. Thus, if the plate tectonic started at 3.8-4.1 Ga (the earliest suggested estimates, 7, 24) the $\delta^{18}\text{O}$ of the CLM were not significantly changed.”

This should clarify it for readers as far as our dataset is concerned.

For the question on slab stacking, there is a new paper in Geology 2022 by Z Wang that may be relevant. Still, the models of Windley, Korenaga, Harrison, Wilde, and others for very early start of plate tectonics seem not considered, but instead take the “consensus” view (which is not always the correct one in science).

By reading an excellent 120-page long 2013 review by Nick Arndt on the topic, the consensus is that 2.7 Ga is the first major zircon peak on zircon frequency-age diagrams, which corresponds to the first major lateral movement on large scale. We also checked the mentioned Wang et al paper, which deals with numerical modeling of eclogite behavior in a particular set of initial and boundary conditions, we need to digest this more before the citation, it is not directly relevant to peridotites.

In response to reviewers, the authors say 2.9-3.0 is Neorchean, but it is not. It is Mesoarchean.

Thank you, changed to Meso

Reviewer 2

The answers to the statistics on when the changes happened, as a simple trend in time,

is a good solution

Reviewer #2 (Remarks to the Author):

Dear Editor,

I am happy with the revisions that were made and would be happy to see this published in its current form.

Best,
Chris Spencer
Kingston, Ontario
19 April 2022

Reviewer 3

The authors say the statistical treatment of the data is improved
The point of further describing the geological differences between the samples with different d_{18O} values is very important. The revised text addresses these concerns as well as possible, for xenolith samples.

Manuscript.

The references are in some place listed as numbers, some by author and year, some both. Please prepare a cleaner manuscript!

>> done

Line 8-85. There is a new paper in PNAS on field examples of Archean eclogitic oceanic crust that may be relevant

We checked but prefer not to add new citation that late in revision especially because it is on eclogites, and also because it takes some time to digest. Our reference list is maxed out.

Line 85-90. Please present the counter-arguments that hotter mantle does not prevent modern style plate tectonics from a physical point of view, see Jun Korenaga papers, or O Weller papers, for their models.

Rephrased a little bit, we would rather not enter the debate not relevant for the flow of the argument in this part of the paper, we cite Korenaga and others already and inserted new reference to Arndt's review paper that covers it all. The discussion is well balanced already.

It seems ref 25 should be moved away from the group at line 93 calling for sagduction (they argued against that) to the references at line 90, and reference 22 be moved to line 26, for the group that argues for sagduction or other things that don't happen on the present Earth.

Done!

I see the questions from reviewers are indeed mostly addressed

Results

The data seem indeed to show a gradual trend over time, with a small deflection focused on the large number of samples (from Udachaya) at 2.0 Ga, and for the post Mesozoic set.

Line 233. Since the Archean samples sampled, are 15% more melt depleted than the post-Archean set., an additional correction is applied to the Archean data. I understand, but still find this troubling, to use the Archean boundary to apply a correction before and after it, and then to look for changes before and after the boundary.

No, this is not done this way, we did not use the age, we used the estimated level of depletion based on chemistry (larger in the Archean) to do the correction. Removed the word "Archean" and "Proterozoic" from this sentence to avoid confusion with age.

Discussion

This is a fair discussion, and has incorporated the comments of the reviewers, so is more balanced than the original version.

Overall the manuscript is improved, and questions answered, to the degree possible with such small changes in the limited data set.

thank you but the dataset is the largest obtained so far for mantle peridotites

Reviewer #3 (Remarks to the Author):

Resubmitted manuscript by Bindeman et al.

The manuscript by Bindeman et al. has now been improved compared to the previous version. My most important concerns with the manuscript have now been dealt with, i.e., the statistical handling of the data was been improved, the supplementary materials were polished, and sufficient context of the sample suite was given in the response letter by the authors. I recommend the work to be published, but noticed a few sentences in the abstract and introduction that need some editing, because they are not fully correct as they are given in the current version of the manuscript:

Line 12 – "Oxygen isotopic ratios are homogenous in the bulk of Earth's mantle". This statement is hard to prove, and recent studies on phenocrysts from ocean island basalts (including high $^3\text{He}/^4\text{He}$ basalts from Iceland) would disagree (work by Maja Rasmussen et al.). Oxygen isotope ratios are pretty homogeneous in mantle xenoliths as is pointed out later in the manuscript; but mantle xenoliths do not necessarily give a representative image of the bulk of Earth's mantle. I'd recommend the authors to tone down this statement.

As this is the Abstract, we just added the word "largely" in front of homogeneous and we think this takes care of the issue. Even the strongest proponents of basaltic magma heterogeneity do not call for more than 1 and a maximum of 2‰ variations and even

this is questioned to originate in the mantle and not as a result of contamination in the crust.

Lines 24 - 26 “d18O of the continental lithospheric mantle has decreased by 0.2‰ since the Archean due to the initiation of plate tectonics and redistribution of low-d18O and high -d18O materials in the subduction zones, explaining crust-upper mantle mass balance“.Why not write that the 0.2‰ change is possibly consistent with crust-upper mantle mass balance? I believe that the causal relationship here is arguable.

Good suggestion, removed “explaining” and added: “... which is consistent with the temporal evolution of the ...”

Lines 29-39: “Earth’s peridotitic mantle is the predominant reservoir of the planet, largely controlling elemental and isotopic mass balances planet-wide”. The statement is incomplete/incorrect. The mantle is the predominant reservoir for lithophile elements, but not for siderophile elements like Fe, Ni, and the HSE; nor for atmophile elements like N and the noble gases. Perhaps write “(..) predominant oxygen reservoir “

Added, “...of lithophilic elements and their isotopes...”

Line 58: “Oxygen is the main element in the solar system” This statement is not correct, because the Si-normalized abundances in the Solar System are much higher for H and He than for O. Perhaps exchange “solar system” with “the terrestrial planets” and the sentence would work.

fair enough. Changed to “...of terrestrial planets”